# A Survey on Transferability of Adversarial Examples Across Deep Neural Networks

**Jindong Gu**[1], **Xiaojun Jia**[2], **Pau de Jorge**[1], **Wenqain Yu**[3], **Xinwei Liu**[4], **Avery Ma**[5],
**Yuan Xun**[4], **Anjun Hu**[1], **Ashkan Khakzar**[1], **Zhijiang Li**[3], **Xiaochun Cao**[6], **Philip Torr**[1]

[1] *Torr Vision Group, University of Oxford, Oxford, United Kingdom*
[2] *Nanyang Technological University, Singapore*
[3] *Wuhan University, Wuhan, China*
[4] *University of Chinese Academy of Sciences, Beijing, China*
[5] *University of Toronto, Toronto, Canada*
[6] *Sun Yat-sen University, Shenzhen, China*

**Reviewed on OpenReview:** *https://openreview.net/forum?id=AYJ3m7BocI*

## Abstract

The emergence of Deep Neural Networks (DNNs) has revolutionized various domains by enabling the resolution of complex tasks spanning image recognition, natural language processing, and scientific problem-solving. However, this progress has also brought to light a concerning vulnerability: adversarial examples. These crafted inputs, imperceptible to humans, can manipulate machine learning models into making erroneous predictions, raising concerns for safety-critical applications. An intriguing property of this phenomenon is the transferability of adversarial examples, where perturbations crafted for one model can deceive another, often with a different architecture. This intriguing property enables "black-box" attacks which circumvents the need for detailed knowledge of the target model. This survey explores the landscape of the adversarial transferability of adversarial examples. We categorize existing methodologies to enhance adversarial transferability and discuss the fundamental principles guiding each approach. While the predominant body of research primarily concentrates on image classification, we also extend our discussion to encompass other vision tasks and beyond. Challenges and opportunities are discussed, highlighting the importance of fortifying DNNs against adversarial vulnerabilities in an evolving landscape.

## 1 Introduction

In recent years, Deep Neural Network (DNN) has evolved as a powerful tool for solving complex tasks, ranging from image recognition (He et al., 2016; Dosovitskiy et al., 2020) and natural language processing (Kenton & Toutanova, 2019; Brown et al., 2020a) to natural science problems (Wang et al., 2023). Since the advent of neural networks, an intriguing and disconcerting phenomenon known as adversarial examples has come into focus (Szegedy et al., 2013; Goodfellow et al., 2014). Adversarial examples are specially crafted inputs that lead machine learning models to make incorrect predictions. These inputs are imperceptibly different from correctly predicted inputs. The existence of adversarial examples poses potential threats to real-world safety-critical DNN-based applications, e.g., medical image analysis (Bortsova et al., 2021) and autonomous driving systems (Kim & Canny, 2017; Kim et al., 2018).

While the existence of adversarial examples has raised concerns about the robustness and reliability of machine learning systems, researchers have uncovered an even more intriguing phenomenon: the *transferability of adversarial examples* (Goodfellow et al., 2014; Papernot et al., 2016). Transferability refers to the ability of an adversarial example designed for one model to successfully deceive a different model, often one with a distinct architecture. With such a property, a successful attack can be implemented without accessing any detail of the target model, such as model architecture, model parameters, and training data.

Table 1: Categorization of transferability-enhancing methods.

| | | |
|---|---|---|
| Optimization-Based | Data Augmentation | Xie et al. (2019); Dong et al. (2019); Lin et al. (2019); Zou et al. (2020); Wu et al. (2021); Li et al. (2020b); Byun et al. (2022); Wang et al. (2021a); Huang & Kong (2022) |
| | Optimization Technique | Goodfellow et al. (2014); Dong et al. (2018); Nesterov (1983); Zou et al. (2022); Wang & He (2021); Xiong et al. (2022); Zhu et al. (2023b); Li et al. (2020a); Qin et al. (2022); Ma et al. (2023); Gubri et al. (2022b) |
| | Loss Objective | Zhang et al. (2022a); Xiao et al. (2021); Li et al. (2020a); Zhao et al. (2021); Fang et al. (2022); Xu et al. (2022b); Li et al. (2023); Qian et al. (2023); Chen et al. (2023a) |
| | Model Component | Zhou et al. (2018); Naseer et al. (2018); Hashemi et al. (2022); Salzmann et al. (2021); Ganeshan et al. (2019); Wang et al. (2021c); Zhang et al. (2022c); Wu et al. (2020b); Inkawhich et al. (2019; 2020b;a); Waseda et al. (2023); Wu et al. (2020a); Guo et al. (2020); Naseer et al. (2022) |
| Generation-Based | Unconditional Generation | Poursaeed et al. (2018); Xiao et al. (2018); Naseer et al. (2019; 2021); Kim et al. (2022); Feng et al. (2022); Zhao et al. (2023) |
| | Class-conditional Generation | Yang et al. (2022); Han et al. (2019); Mao et al. (2020); Han et al. (2019); Phan et al. (2020); Chen et al. (2023b;d; 2024) |

The recent surge in interest surrounding the transferability of adversarial examples is attributed to its potential application in executing black-box attacks (Papernot et al., 2016; Liu et al., 2017). Delving into the reasons behind the capacity of adversarial examples tailored for one model to deceive others provides researchers with an opportunity to acquire a profound comprehension of the fundamental mechanisms underpinning the susceptibility of DNNs (Wu & Zhu, 2020). Moreover, a comprehensive understanding of transferability offers the possibility of fostering the creation of adversarially robust models, capable of effectively countering adversarial attacks (Jia et al., 2022b; Waseda et al., 2023; Ilyas et al., 2019).

Given the growing volume of publications on adversarial examples, several survey papers (Sun et al., 2018; Zhang & Li, 2019; Wiyatno et al., 2019; Serban et al., 2020; Han et al., 2023b) have emerged, seeking to encapsulate these phenomena from diverse viewpoints. Yet, a comprehensive survey specifically addressing the transferability of adversarial examples remains absent. This study endeavors to bridge that gap, embarking on an extensive investigation into the realm of adversarial transferability. To this end, we thoroughly review the latest research on transferability assessment, methods to enhance transferability, and the associated challenges and prospects. Specifically, as shown in Table 1, we categorize the current transferability-enhancing methods into two major categories: (1) **optimization-based** methods where one directly optimizes for the adversarial perturbations based on one or more surrogate models at inference time, without introducing additional generative models, and (2) **generation-based** methods that introduce generative models dedicated for adversary synthesis. Moreover, we also examine adversarial transferability beyond the commonly studied misclassification attacks and provide a summary of such phenomenon in other tasks (e.g. image retrieval, object detection, segmentation, etc.). Upon assessing the current advancements in adversarial transferability research, we then outline a few challenges and potential avenues for future investigations.

The organization of this paper is as follows: Section 2 first provides the terminology and mathematical notations used across the paper, and then introduces the formulation and evaluation metrics of adversarial transferability. Sections 3-5 present various techniques to improve the adversarial transferability of adversarial examples. Section 3 examines transferability-enhancing methods that are applicable to optimisation-based transferable attacks. These techniques are categorized into four perspectives: data processing, optimization, loss objective, and model architectures. In Section 4, various generation-based transferable attacks are presented. Section 5 describes the research on adversarial transferability beyond image classification. Concretely, transferability-enhancing techniques in various vision tasks, natural language processing tasks as well as the ones across tasks are discussed. The current challenges and future opportunities in adversarial transferability are discussed in Section 6. The last section concludes our work.

In order to facilitate the literature search, we also built and released a project page where the related papers are organized and listed[1]. The page will be maintained and updated regularly. As the landscape of DNN continues to evolve, understanding the intricacies of adversarial examples and their transferability is of paramount importance. By shedding light on these vulnerabilities, we hope to contribute to the development of more secure and robust DNN models that can withstand the challenges posed by adversarial attacks.

## 2 Preliminary

In this section, we first introduce the terminology and mathematical notations used across the paper. Then, we introduce the formulation of adversarial transferability. In the last part, the evaluation of adversarial transferability is presented.

### 2.1 Terminology and Notations

The terminologies relevant to the topic of adversarial transferability and mathematical annotations are listed in Tab. 2 and Tab. 3, respectively.

Table 2: The used terminologies are listed. They are followed across the paper.

| | |
|---|---|
| *adversarial perturbation* | A small artificial perturbation that can cause misclassification of a neural network when added to the input image. |
| *target model* | The target model (e.g. deep neural network) to attack. |
| *surrogate model* | The model built to approximate the target model for generating adversarial examples. |
| *white / black-box attacks* | White attacks can access the target model (e.g. architecture and parameters), while black-box attacks cannot. |
| *untargeted/targeted attack* | The goal of untargeted attacks is to cause misclassifications of a target model, while targeted attacks aim to mislead the target model to classify an image into a specific target class. |
| *clean accuracy* | The model performance on the original clean test dataset. |
| *fooling rate* | The percentage of images that are misclassified the target model. |

Table 3: The used mathematical notations are listed. They are followed across the paper.

| | | | |
|---|---|---|---|
| $x$ | A clean input image | $y$ | A ground-truth label of an image |
| $\delta$ | Adversarial perturbation | $\epsilon$ | Range of adversarial perturbation |
| $\chi$ | Input distribution | $x^{adv}$ | Adversarial example $x + \delta$ of the input $x$ |
| $y^t$ | Target class of an adversarial attack | $x^{adv(t)}$ | Adversarial input at the end of $t^{th}$ iteration |
| $f_t$ | Target model under attack | $f_s$ | Surrogate (source) model for AE creation |
| $f^i(x)$ | the $i^{th}$ model output probability for the input $x$ | $H_k^l$ | $k^{th}$ activation in the $l^{th}$ layer of the target model |
| $H^l$ | the $l^{th}$ layer of the target network | $z^i$ | Model output logits |
| $AE$ | Adversarial Example | $AT$ | Adversarial Transferability of AE |
| $Acc$ | Clean accuracy on clean dataset | $FR$ | Fooling rate on target model |

---

[1]https://github.com/JindongGu/awesome_adversarial_transferability

## 2.2 Formulation of Adversarial Transferability

Given an adversarial example $x^{adv}$ of the input image $x$ with the label $y$ and two models $f_s(\cdot)$ and $f_t(\cdot)$, adversarial transferability describes the phenomenon that the adversarial example that is able to fool the model $f_s(\cdot)$ can also fooling another model $f_t(\cdot)$. Formally speaking, the adversarial transferability of untargeted attacks can be formulated as follows:

$$\text{argmax}_i f_t^i(x^{adv}) \neq y, \text{given argmax}_i f_s^i(x^{adv}) \neq y \tag{1}$$

Similarly, the targeted transferable attacks can be described as:

$$\text{argmax}_i f_t^i(x^{adv}) = y^t, \text{given argmax}_i f_s^i(x^{adv}) = y^t \text{ and argmax}_i f_s^i(x) = y \tag{2}$$

The process to create $x^{adv}$ for a given example $x$ is detailed in Sec. 3 and Sec. 4.

## 2.3 Evaluation of Adversarial Transferability

**Fooling Rate (FR).** The most popular metric used to evaluate adversarial transferability is FR. We denote $P$ as the number of adversarial examples that successfully fool a source model. Among them, the number of examples that are also able to fool a target model is $Q$. The Fooling Rate is then defined as $\text{FR} = \frac{Q}{P}$. The higher the FR is, the more transferable the adversarial examples are.

**Interest Class Rank (ICR).** According to Zhang et al. (2022a), an interest class is the ground-truth class in untargeted attacks or the target class in targeted attacks. FR only indicates whether the interest class ranks top-1, which may not be an in-depth indication of transferability. To gain deeper insights into transferability, it is valuable to consider the ICR, which represents the rank of the interest class after the attack. In untargeted attacks, a higher ICR indicates higher transferability, while in targeted attacks, a higher ICR suggests lower transferability.

**Knowledge Transfer-based Metrics.** Liang et al. (2021) considered all potential adversarial perturbation vectors and proposed two practical metrics for transferability evaluation. The transferability of dataset $x \sim D$ from source model $f_s$ to target model $f_t$ are defined as follows:

$$\alpha_1^{f_s \to f_t}(x) = \frac{\ell(f_t(x), f_t(x + \delta_{f_s, \varepsilon}(x)))}{\ell(f_t(x), f_t(x + \delta_{f_t, \varepsilon}(x)))} \tag{3}$$

where $\delta_{f_s, \varepsilon}(x)$ is the adversrial perturbation generated on surrogate model $f_s$ while $\delta_{f_t, \varepsilon}(x)$ is that on target model $f_t$. The first metric $\alpha_1$ measures the difference between two loss functions which can indicate the performance of two attacks: black-box attack from $f_s$ to $f_t$ and white-box attack on $f_t$. Tranferabilty from $f_s$ to $f_t$ is high when $\alpha_1$ is high.

$$\alpha_2^{f_s \to f_t} = \left\| \mathbb{E}_{x \sim D}[\widehat{\Delta_{f_s \to f_s}}(x) \cdot \widehat{\Delta_{f_s \to f_t}}(x)^\top] \right\|_F \tag{4}$$

where

$$\Delta_{f_s \to f_s}(x) = f_s(x + \delta_{f_s, \varepsilon}(x)) - f_s(x), \quad \Delta_{f_s \to f_t}(x) = f_t(x + \delta_{f_s, \varepsilon}(x)) - f_t(x) \tag{5}$$

$\widehat{\cdot}$ operation denotes the corresponding unit-length vector and $\|\cdot\|_F$ denotes the Frobenius norm. The second metric $\alpha_2$ measures the relationship between two deviation directions, indicating white-box attacks on $f_s$ and black-box attacks from $f_s$ to $f_t$ respectively.

Liang et al. (2021) argue that these two metrics represent complementary perspectives of transferability: $\alpha_1$ represents how often the adversarial attack transfers, while $\alpha_2$ encodes directional information of the output deviation.

## 3 Optimization-Based Transferable Attacks

In this section, we introduce optimization-based transferability-enhancing methods: a class of methods that seeks adversarial perturbation at test time based on one or more surrogate models without introducing or

training additional models (e.g. a generative model). Based on our formulation in Section 2.2, the process to obtain transferable adversarial perturbations with this class of methods can be expressed as:

$$\delta^* = \arg\max_\delta \mathbb{E}_T \ \ell(f_s(T(x + \delta)), y), \ \ s.t. \ ||\delta||_\infty \leq \epsilon \tag{6}$$

where $\delta$ is an adversarial perturbation of the same size as the input, $T(\cdot)$ is data augmentation operations, $\ell(\cdot)$ is a designed loss, and $f_s(\cdot)$ is a model used in the optimization process, which could be slightly modified version of a surrogate model. The examples of the modifications are linearizing the surrogate model by removing the non-linear activation functions and highlighting skip connections in backpropagation passes.

The problem in Equation 6 is approximately solved with Projected Gradient Descent (Madry et al., 2017). Multi-step attacks (e.g. (Madry et al., 2017)) update the perturbation iteratively as follows

$$\delta^{(t+1)} = \text{Clip}^\epsilon(\delta^t + \alpha \cdot \text{sign}(\nabla_x \ell)), \tag{7}$$

where $\alpha$ is the step size to update the perturbation, $x^{adv(t+1)} = x + \delta^t$, and $\text{Clip}^\epsilon(\cdot)$ is a clipping function to make the constraint $||\delta||_\infty \leq \epsilon$ satisfied. In contrast, single-step attacks (e.g. (Szegedy et al., 2013; Goodfellow et al., 2014)) update the adversarial perturbation with only one attack iteration.

Given the expression in Equation 6, we categorize the transferability-enhancing methods into four categories: data augmentation-based, optimization-based, model-based, and loss-based.

### 3.1 Data Augmentation-Based Transferability Enhancing Methods

The family of methods discussed in this section, referred to as Data Augmentation-based methods, are all based on the rationale of applying data augmentation strategies. In these strategies, when computing the adversaries on the surrogate model $f_s$, an input transformation $T(\cdot)$ with a stochastic component is applied to increase adversarial transferability by preventing overfitting to the surrogate model. In the following, we will discuss the specific instance of $T(\cdot)$ for each method.

**Diverse Inputs Method (DIM).** Xie et al. (2019) are the first to suggest applying differentiable input transformations to the clean image. In particular, their DIM consists of applying a different transformation at each iteration of multi-step adversarial attacks. They perform random resizing and padding with a certain probability $p$, that is:

$$T(x) = \begin{cases} x & \text{with probability } 1 - p \\ \text{pad}(\text{resize}(x)) & \text{with probability } p \end{cases} \tag{8}$$

and as $p$ increases, the transferability of iterative attacks improves most significantly. They also notice that although this is tied to a drop in the white-box performance, the latter is much milder.

**Translation Invariance Method (TIM).** Dong et al. (2019) study the *discriminative image regions* of different models, i.e. the image regions more important to classifiers output. They observe different models leverage different regions of the image, especially if adversarially trained. This motivates them to propose the *Translation Invariance Method* (TIM). That is, they want to compute an adversarial perturbation that works for the original image and any of its translated versions (where the position of all pixels is shifted by a fixed amount), therefore:

$$T(x)[i, j] = x[i + t_x, j + t_y] \tag{9}$$

where $t_x$ and $t_y$ define the shift. Moreover, Dong shows that such perturbations can be found with little extra cost by applying a kernel matrix on the gradients. This method can be combined with other attacks or augmentation techniques (e.g. DIM) to further improve transferability.

**Scale Invariance Method (SIM).** When attacking in a black-box setting, Dong et al. (2018) showed that computing an attack for an ensemble of models improves transferability albeit at an increased computational cost. Motivated by this fact, Lin et al. (2019) introduce the concept of *loss-preserving transformation* (any input transformation $T$ that satisfies $\ell(f(T(x)), y) \approx \ell(f(x), y) \ \forall x$) and of *model augmentation* (given a loss-preserving transformation ($T$), then $f' = f(T(\cdot))$ would be an augmented model of $f$). One can then

use different model augmentations and treat them as an ensemble. In particular, Lin et al. (2019) find downscaling the input image tends to preserve the loss,

$$T(x, i) = S_i(x) = x/2^i, \tag{10}$$

where $S_i(x)$ scales the pixels of x and $i$ is the number of predefined scales. In a similar fashion as TIM, the authors propose the Scale Invariance Method (SIM) to find perturbations that work for any downscaled version of the input. Note that this is different from DIM since SIM optimizes the perturbation for different scales at once while DIM applies a different single resizing and padding at each gradient ascent step.

**Resized Diverse Inputs (RDI).** Zou et al. (2020) Introduce a variant of DIM where the inputs are resized back to the original size after applying DIM augmentations, i.e. *Resized Diverse Inputs* (RDI). Resizing the inputs allows them to test much more aggressive resizing and padding augmentations which boost performance. Similarly to SIM, they also propose to ensemble the gradients from different inputs, however, instead of multiple-scale images, they use RDI augmentations with varying strength. They also observe that keeping the attack optimizer step-size $\alpha = \epsilon$ constant, further improves the success rate of the attacks.

$$T(x) = \text{resize}(\text{resize}(x, H/s, W/s), H, W) \tag{11}$$

**Adversarial Transformation Transfer Attack (ATTA).** A common theme in previous methods has been that combining multiple image transformations (DIM + TIM + SIM) usually leads to better results than using just one set of augmentations. However, all previous methods have focused on a fixed set of augmentations, which limits the diversity of augmentations even if combined. Wu et al. (2021) introduce the *Adversarial Transformation Transfer Attack* where input transformations are parametrized by a network that is optimized alongside the adversarial perturbations to minimize the impact of adversarial attacks. Thus, improving the resilience of the final attacks to input perturbations.

$$T(x) = \psi_\theta(x), \quad \theta = \arg\min_\theta \ell(f_s(\psi_\theta(x+\delta)), y) \tag{12}$$

**Regionally Homogeneous Perturbations (RHP).** Li et al. (2020b) observe that adversarial perturbations tend to have high regional homogeneity (i.e. gradients corresponding to neighboring pixels tend to be similar), especially when models are adversarially robust. Based on this observation they propose to apply a parametrized normalization layer to the computed gradients that encourages *Regionally Homogeneous Perturbations*(RHP). Their objective can be written as:

$$x^{adv} = x + \epsilon \cdot \text{sign}(T(\nabla_x \ell(f_s(x), y))), \tag{13}$$

where $T = \phi_\theta(\cdot)$ is a parametrized transformation on the gradients rather than the input. This transformation is optimized to maximize the loss of the perturbed sample $x^{adv}$. Interestingly, they observe that when optimizing the normalization parameters on a large number of images, the generated perturbations converge to an input-independent perturbation. Hence, although not explicitly enforced, they find RHP leads to universal perturbation (i.e. perturbations that can fool the model for many different inputs).

**Object-based Diverse Input (ODI).** Motivated by the ability of humans to robustly recognize images even when projected over 3D objects (e.g. an image printed on a mug, or a t-shirt), Byun et al. (2022) present a method named *Object-based Diverse Input* (ODI), a variant of DIM which renders images on a set of 3D objects when computing the perturbations as a more powerful technique to perform data augmentation technique.

$$T(x) = \Pi(x, O), \tag{14}$$

where $\Pi$ is a projection operation onto the surface of a 3D mesh and $O$ represents the 3D object.

**Admix.** Inspired by the success of Mixup (training models with convex combinations of pairs of examples and their labels) in the context of data augmentation for classification model training (Zhang et al., 2017), Wang et al. (2021a) study this technique in the context of fostering transferability of adversarial example. However, they find that mixing the labels of two images significantly degrades the white-box performance of adversarial attacks and brings little improvement in terms of transferability. Hence, they present a variation

of Mixup (Admix) where a small portion of different randomly sampled images is added to the original one, but the label remains unmodified. This increases the diversity of inputs, and thus improves transferability but does not harm white-box performance. In this case,

$$T(x) = \eta x + \tau x', \text{ where } \eta < 1 \text{ and } \tau < \eta. \tag{15}$$

The blending parameters $\tau, \eta$ are randomly sampled and the restriction $\tau < \eta$ ensures the resulting image does not differ too much from the original one. The additional images $x'$ are randomly sampled from other categories.

**Transferable Attack based on Integrated Gradients (TAIG).** Huang & Kong (2022) leverage the concept of integrated gradients (i.e. a line integral of the gradients between two images) introduced by Sundararajan et al. (2017a). Instead of optimizing the attack based on the sign of the gradients (e.g. with FGSM) they use the sign of the integrated gradients from the origin to the target image. Moreover, they show that following a piece-wise linear random path improves results further. We can formalize their objective as follows:

$$\tilde{x} = x + \alpha \cdot \text{sign}(\text{IG}(f_s, x, x')), \tag{16}$$

and $\text{IG}(f_s, x, x')$ is the integrated gradient between $x$ and another image $x'$.

### 3.2 Optimization Technique-Based Transferability Enhancing Methods

The generation of transferable adversarial examples can be formulated as an optimization problem of Equation 6. In the last section, we presented how the input data augmentations influence the transferability of the created adversarial perturbations. In this section, we describe how the current work improves adversarial transferability from the perspective of the optimization technique itself.

In this section, we focus on perturbations constrained by an $\ell_\infty$ ball with radius $\epsilon$, that is, $||x^{adv} - x||_\infty \le \epsilon$. To understand the rest of this section, we begin by formalizing the iterative variant of the fast gradient sign method (I-FGSM) (Goodfellow et al., 2014), which serves as the basis for the development of other methods. The I-FGSM has the following update rule:

$$g^{(t+1)} = \nabla\ell(x^{adv(t)}, y),$$
$$x^{adv(t+1)} = \text{Clip}_x^\epsilon\{x^{adv(t)} + \alpha \cdot \text{sign}(g^{(t+1)})\}, \tag{17}$$

where $g^{(t)}$ is the gradient of the loss function with respect to the input, $\alpha$ denotes the step size at each iteration, and $\text{Clip}_x^\epsilon$ ensures that the perturbation satisfies the $\ell_\infty$-norm constraints.

**Momentum (MI-FGSM).** One of the simplest and most widely used techniques to improve the generalizability of neural networks is to incorporate momentum in the gradient direction (Polyak, 1964; Duch & Korczak, 1998). Motivated by this, Dong et al. (2018) propose a momentum iterative fast gradient sign method (MI-FGSM) to improve the vanilla iterative FGSM methods by integrating the momentum term in the input gradient. At each iteration, MI-FGSM updates $g^{(t+1)}$ by using

$$g^{(t+1)} = \mu \cdot g^{(t)} + \frac{\nabla\ell(x^{adv(t)}, y)}{||\nabla\ell(x^{adv(t)}, y)||_1}, \tag{18}$$

where $g^{(t)}$ now represents the accumulated gradients at iteration $t$, $\mu$ denotes the decay factor of $g^{(t)}$, and the formulation for $x^{adv(t+1)}$ remains the same as (17). By integrating the momentum term into the iterative attack process, MI-FGSM can help escape from poor local maxima, leading to higher transferability for adversarial examples.

More variants of I-FGSM have been proposed to make the created adversarial perturbation more transferable. Similar to the development of the DNN optimization techniques, the first-order moment, the second-order moment, and more components are integrated successively to improve adversarial transferability. Those variants include Nesterov (NI-FGSM) (Lin et al., 2019), Adam (AI-FGTM) (Zou et al., 2022), and Variance Tuning (VNI/VMI-FGSM) (Wang & He, 2021), the details of which can be found in Appendix A.

**Stochastic Variance Reduced Ensemble (SVRE).** Xiong et al. (2022) propose a variance-tuning strategy to generate transferable adversarial attacks. Conventional ensemble-based approaches leverage gradients from multiple models to increase the transferability of the perturbation. Xiong et al. (2022) analogize this process as a stochastic gradient descent optimization process, in which the variance of the gradients on different models may lead to poor local optima. As such, they propose to reduce the variance of the gradient by using an additional iterative procedure to compute an unbiased estimate of the gradient of the ensemble. SVRE can be generalized to any iterative gradient-based attack.

**Gradient Relevance Attack (GRA).** Zhu et al. (2023b) introduce GRA, an attack strategy built upon MI-FGSM and VMI-FGSM. This approach incorporates two key techniques to further increase transferability. First, during (18), the authors notice that the sign of the perturbation fluctuates frequently. Given the constant step size in the iterative process, this could suggest the optimization getting trapped in local optima. To address this, they introduced a decay indicator, adjusting the step size in response to these fluctuations, thus refining the optimization procedure. Additionally, they argue that in VMI-FGSM, the tuning strategy used during the last iteration might not accurately represent the loss variance at the current iteration. Therefore, they propose to use dot-product attention (Vaswani et al., 2017) to compute the gradient relevance between $x^{adv(t)}$ and its neighbors, providing a more precise representation of the regional information surrounding $x^{adv(t)}$. This relevance framework is then used to fine-tune the update direction.

**Adaptive Gradient Method (AGM).** While the previous methods focus on improving the optimization algorithm, another angle of attack is through the optimization objective. Li et al. (2020a) demonstrate that the cross-entropy loss, commonly utilized in iterative gradient-based methods, is not suitable for generating transferable perturbations in the targeted scenario. Analytically, they demonstrate the problem of vanishing gradients when computing the input gradient with respect to the cross-entropy loss. Although this problem is circumvented by projecting the gradient to a unit $\ell_1$ ball, they empirically show in the targeted setting that this normalized gradient is overpowered by the momentum accumulated from early iterations. Notice that because historical momentum dominates the update, the effect of the gradient at every iteration diminishes, and thus the update is not optimal in finding the direction toward the targeted class. To circumvent the vanishing gradient problem, they propose to incorporate the Poincaré metric into the loss function. They empirically demonstrate that the input gradient with respect to the Poincaré metric can better capture the relative magnitude between the gradient magnitude and the distance from the perturbed data point to the target class, leading to more effective iterative updates during the attack process.

**Reverse Adversarial Perturbation (RAP).** Improving the adversarial robustness of neural networks by training with perturbed examples has been studied under the robust optimization framework, which is also known as a min-max optimization problem. Similarly, Qin et al. (2022) propose to improve the transferability of adversarial examples under such a min-max bi-level optimization framework. They introduce RAP that encourages points in the vicinity of the perturbed data point to have similar high-loss values. Unlike the conventional I-FGSM formulation, the inner-maximization procedure of RAP first finds perturbations with the goal of minimizing the loss, whereas the outer-minimization process updates the perturbed data point to find a new point added with the provided reverse perturbation that leads to a higher loss.

**Momentum Integrated Gradients (MIG).** The integrated gradient (IG) is a model-agnostic interpretability method that attributions the prediction of a neural network to its inputs (Sundararajan et al., 2017b). The gradient derived from this method can be understood as saliency scores assigned to the input pixels. Notably, a distinct feature of IG is its implementation invariance, meaning that the gradients only depend on the input and output of the model and are not affected by the model structure. Such a characteristic can be especially beneficial when improving the transferability of perturbation across different model architectures. Inspired by these, Ma et al. (2023) propose MIG which incorporates IG in the MI process.

### 3.3 Loss Objective-Based Transferability Enhancing Methods

The loss objective used to create adversarial examples on the surrogate models is the cross-entropy loss in Equation 6. Various designs have also been explored to improve the transferability of the created adversarial examples.

**Normalized CE Loss.** To increase the attack strength, Zhang et al. (2022a) identify that the weakness of the commonly used losses lies in prioritizing the speed to fool the network instead of maximizing its strength. With an intuitive interpretation of the logit gradient from the geometric perspective, they propose a new normalized CE loss that guides the logit to be updated in the direction of implicitly maximizing its rank distance from the ground-truth class. For boosting the top-k strength, the loss function consists of two parts: the commonly used CE, and a normalization part, averaging the CE calculated for each class. The loss function they term Relative CE loss or RCE loss in short is formulated as follows:

$$\text{RCE}\left(x^{adv(t)}, y\right) = \text{CE}\left(x^{adv(t)}, y\right) - \frac{1}{K} \sum_{k=1}^{K} \text{CE}\left(x^{adv(t)}, y_k\right) \tag{19}$$

Their proposed RCE loss in the equation above achieved a strong top-k attack in both white-box and transferable black-box settings.

**Generative Model as Regularization.** Focusing on black-box attacks in restricted-access scenarios, Xiao et al. (2021) propose to generate transferable adversarial patches (TAPs) to evaluate the robustness of face recognition models. Some adversarial attacks based on transferability show that certain adversarial examples of white-box alternative models $g$ can be maintained adversarial to black-box target models $f$. Suppose $g$ is a white-box face recognition model that is accessible to the attacker, the optimization problem to generate the adversarial patch on the substitute model can be described as follows:

$$\max_{\mathbf{x}} \mathcal{L}_g\left(\mathbf{x}, \mathbf{x}_t\right), \ \text{s.t.} \ \mathbf{x} \odot (1 - \mathbf{M}) = \mathbf{x}_s \odot (1 - \mathbf{M}), \tag{20}$$

where $\mathcal{L}_g$ is a differentiable adversarial objective, $\odot$ is the element-wise product, and $M \in \{0, 1\}^n$ is a binary mask. However, when solving this optimization problem, even the state-of-the-art optimization algorithms are still struggling to get rid of local optima with unsatisfactory transferability. To solve this optimization challenge, Xiao et al. (2021) propose to optimize the adversarial patch on a low-dimensional manifold as a regularization. Since the manifold poses a specific structure on the optimization dynamics, they consider a good manifold should have two properties: sufficient capacity and well regularized. In order to balance the requirements of capacity and regularity, they learn the manifold using a generative model, which is pre-trained on natural human face data and can combine different face features by manipulating latent vectors to generate diverse, unseen faces, e.g., eye color, eyebrow thickness, etc. They propose to use the generative model to generate adversarial patches and optimize the patches by means of latent vectors. Thus the above optimization problem is improved as:

$$\max_{\mathbf{s} \in \mathcal{S}} \mathcal{L}_g\left(\mathbf{x}, \mathbf{x}_t\right),$$
$$\text{s.t.} \ \mathbf{x} \odot (1 - \mathbf{M}) = \mathbf{x}_s \odot (1 - \mathbf{M}),$$
$$\mathbf{x} \odot \mathbf{M} = h(\mathbf{s}) \odot \mathbf{M}, \tag{21}$$

where the second constrain restricts the adversarial patch on the low-dimensional manifold represented by the generative model, $h(s) : S \to \mathbb{R}^n$ denote the pre-trained generative model and $S$ is its latent space. When constrained on this manifold, the adversarial perturbations resemble face-like features. For different face recognition models, they expect that the responses to the face-like features are effectively related, which will improve the transferability of the adversarial patches.

**Metric Learning as Regularization.** Most of the previous research on transferability has been conducted in non-targeted contexts. However, targeted adversarial examples are more difficult to transfer than non-targeted examples. Li et al. (2020a) find that there are two problems that can make it difficult to produce transferable examples: (1) During an iterative attack, the size of the gradient gradually decreases, leading to excessive consistency between two consecutive noises during momentum accumulation, which is referred to as noise curing. (2) Targeted adversarial examples must not only approach the target class but also stray from the ground-truth class. To overcome these two problems, they discard the Euclidean standard space and introduce Poincaré ball as a metric space for the first time to solve the noise curing problem, which makes the magnitude of gradient adaptive and the noise direction more flexible during the iterative attack

process. Instead of the traditional cross-entropy loss, they propose Poincaré Distance Metric loss $\mathcal{L}_{Po}$, which makes the gradient increase only when it is close to the target class. Since all the points of the Poincaré ball are inside a n-dimensional unit $\ell_2$ ball, the distance between two points can be defined as:

$$d(u, v) = \operatorname{arccosh}(1 + \delta(u, v)), \tag{22}$$

where $u$ and $v$ are two points in n-dimensional Euclid space $\mathbb{R}^n$ with $\ell_2$ norm less than one, and $\delta(u, v)$ is an isometric invariant defined as follow:

$$\delta(u, v) = 2\frac{\|u - v\|^2}{(1 - \|u\|^2)(1 - \|v\|^2)}, \tag{23}$$

However, the fused logits are not satisfied $\|l(x)\|_2 < 1$, so they normalize the logits by the $\ell_1$ distance. And they subtract the one hot target label $y$ from a small constant $\xi = 0.0001$ because the distance from any point to $y$ is $+\infty$. Thus, the final Poincar´e distance metric loss can be described as follows:

$$\ell_{Po}(x, y) = d(u, v) = \operatorname{arccosh}(1 + \delta(u, v)), \tag{24}$$

where $u = l_k(x)/\|l_k(x)\|_1$, $v = max\{y - \xi, 0\}$, and $l(x)$ is the fused logits.

In targeted attacks, the loss function usually only considers the desired target label. But sometimes, the generated adversarial examples are too similar to the original class, causing some to still be classified correctly by the target model. Therefore they also utilize the real label information as an addition during the iterative attack, using triplet loss, to help the adversarial examples stay away from the real labels to get better transferability. They use the logits of clean images $l(x_{clean})$, one-hot target label and true label $y_{tar}$, $y_{true}$ as the triplet input:

$$\ell_{trip}(y_{tar}, l(x_i), y_{true}) = [D(l(x_i), y_{tar}) - D(l(x_i), y_{true}) + \gamma]_+. \tag{25}$$

Since the $l(x^{adv})$ is not normalized, so they use the angular distance $D(\cdot)$ as a distance metric, which eliminates the influence of the norm on the loss value. The distance calculation can be described as follows:

$$D(l(x^{adv}), y_{tar}) = 1 - \frac{|l(x^{adv}) \cdot y_{tar}|}{\|l(x^{adv})\|_2 \|y_{tar}\|_2}. \tag{26}$$

Therefore, after adding the triplet loss term, their overall loss function:

$$\ell_{all} = \ell_{Po}(l(x), y_{tar}) + \lambda \cdot \ell_{trip}(y_{tar}, l(x_i), y_{true}). \tag{27}$$

**Simple Logit Loss.** Zhao et al. (2021) review transferable targeted attacks and find that their difficulties are overestimated due to the blind spots in traditional evaluation procedures since current works have unreasonably restricted attack optimization to a few iterations. Their study shows that with enough iterations, even conventional I-FGSM integrated with simple transfer methods can easily achieve high targeted transferability. They also demonstrate that attacks utilizing simple logit loss can further improve the targeted transferability by a very large margin, leading to results that are competitive with state-of-the-art techniques. The simple logit loss can be expressed as:

$$\ell = \max_{\mathbf{x}'} Z_t(x'), \tag{28}$$

where $Z_t(\cdot)$ denotes the logit output before the softmax layer with respect to the target class.

**Meta Loss.** Instead of simply generating adversarial perturbations directly in these tasks, Fang et al. (2022) optimize them using meta-learning concepts so that the perturbations can be better adapted to various conditions. The meta-learning method is implemented in each iteration of the perturbation update. In each iteration, they divide the task into a support set $\mathcal{S}$ and a query set $\mathcal{Q}$, perform meta-training (training on the support set) and meta-testing (fine-tuning on the query set) multiple times, and finally update the

adversarial perturbations. In each meta-learning iteration, they sample a subset $\mathcal{S}_i \in \mathcal{S}$ and calculate the average gradient with respect to input as:

$$\boldsymbol{g}_{spt} = \frac{1}{|\mathcal{S}_i|} \sum_{(\boldsymbol{x}_s, \boldsymbol{\gamma}_s) \in \mathcal{S}_i} G(\boldsymbol{x}_s, \gamma_s), \tag{29}$$

where $G$ denotes the gradient updation of adversarial perturbations as in I-FGSM: $G(\boldsymbol{x}_{adv}, f) = \nabla_{\boldsymbol{x}_{adv}} \mathcal{L}(f(\boldsymbol{x}_{adv}), y)$. The $\boldsymbol{\gamma} = [\gamma_1, \gamma_2, ..., \gamma_L]^T \in [0, 1]^T$ denotes the set of decay factors during model augmentation, and the factor $\gamma_i$ defaults for $i$-th residual layer. Since the optimization of $\boldsymbol{\gamma}$ represents the augmentation of the model $f$, for ease of writing, they replaced $\boldsymbol{\gamma}$ with $f$, i.e, $G(\boldsymbol{x}_{adv}, \gamma_s) = G(\boldsymbol{x}_{adv}, f) = \nabla_{\boldsymbol{x}_{adv}} \mathcal{L}(f(\boldsymbol{x}_{adv}), y)$.

Then, similar to FGSM, they obtain the temporary perturbation with a single-step update:

$$\boldsymbol{\delta}' = \epsilon \cdot \text{sign}(\boldsymbol{g}_{spt}). \tag{30}$$

Then they finetune on the query set $\mathcal{Q}$ and compute the query gradient $\boldsymbol{g}_{qry}$ by adding the temporary perturbation so that it can adapt more tasks:

$$\boldsymbol{g}_{qry} = \frac{1}{|\mathcal{Q}|} \sum_{(\boldsymbol{x}_q, \boldsymbol{\gamma}_q) \in \mathcal{Q}} G(\boldsymbol{x}_q + \delta', \gamma_q). \tag{31}$$

Finally, they update the actual adversarial perturbation with the gradient from both the support set and the query set for maximum utilization:

$$\boldsymbol{x}_{adv}^{t+1} = \Pi_\epsilon^{\boldsymbol{x}} \left( \boldsymbol{x}_{adv}^t + \alpha \cdot \text{sign} \left( \overline{\boldsymbol{g}}_{spt} + \overline{\boldsymbol{g}}_{qry} \right) \right) \tag{32}$$

where $\overline{\boldsymbol{g}}_{spt}$ and $\overline{\boldsymbol{g}}_{qry}$ denote the average gradient over meta-learning iterations, respectively.

**Domain transferability Through Regularization.** Xu et al. (2022b) propose a theoretical framework to analyze the sufficient conditions for domain transferability from the view of function class regularization. They prove that shrinking the function class of feature extractors during training monotonically decreases a tight upper bound on the relative domain transferability loss. Therefore, it is reasonable to expect that imposing regularization on the feature extractor during training can lead to better relative domain transferability.

**More Bayesian Attack.** Many existing works enhance attack transferability by increasing the diversity in inputs of some substitute models. Li et al. (2023) propose to attack a Bayesian model for achieving desirable transferability. By introducing probability measures for the weights and biases of the alternative models, all these parameters can be represented under the assumption of some to-be-learned distribution. In this way, an infinite number of ensembles of DNNs (which appear to be jointly trained) can be obtained in a single training session. And then by maximizing the average predictive loss of this model distribution, adversarial examples are produced, which is referred to as the posterior learned in a Bayesian manner. The attacks performing on the ensemble of the set of $M$ models can be formulated as:

$$\underset{\|\boldsymbol{\Delta}\mathbf{x}\|_p \leq \epsilon}{\arg\min} \frac{1}{M} \sum_{i=1}^M p(y \mid \mathbf{x} + \boldsymbol{\Delta}\mathbf{x}, \mathbf{w}_i) = \underset{\|\boldsymbol{\Delta}\mathbf{x}\|_p \leq \epsilon}{\arg\max} \frac{1}{M} \sum_{i=1}^M L(\mathbf{x} + \boldsymbol{\Delta}\mathbf{x}, y, \mathbf{w}_i), \text{ s.t. } \mathbf{w}_i \sim p(\mathbf{w} \mid \mathcal{D}), \tag{33}$$

where $L(\cdot, \cdot, \boldsymbol{w}_i)$ is a function evaluating prediction loss of a DNN model parameterized by $\boldsymbol{w}_i$. Using iterative optimization methods, different sets of models can be sampled at different iteration stages, as if an infinite number of substitute models existed.

**Lightweight Ensemble Attack (LEA).** Qian et al. (2023) notice three models with non-overlapping vulnerable frequency regions that can cover a sufficiently large vulnerable subspace. Based on this finding, they propose LEA2, a lightweight ensemble adversarial attack consisting of standard, weakly robust, and robust models. Furthermore, they analyze Gaussian noise from a frequency view and find that Gaussian

noise occurs in the vulnerable frequency regions of standard models. As a result, they replace traditional models with Gaussian noise to ensure that high-frequency vulnerable regions are used while lowering attack time consumption. They first define the vulnerable frequency regions and the adversarial example $x'$ is generated by the following optimization:

$$\arg\max_{\delta} -\log\left(\left(\sum_{i=1}^{M_1} w_i S_{\text{robust}}^i (x+r+\delta) + \sum_{j=1}^{M_2} w_j S_{\text{weak}}^j (x+r+\delta)\right) \cdot \mathbf{1}_y\right), \tag{34}$$

where $r \sim N(0, \sigma^2)$ is the Gaussian noise, $M_1$ and $M_2$ are the number of robust models and weakly robust models respectively, $S_{robust}$ and $S_{weak}$ represent the corresponding softmax outputs, $\mathbf{1}_y$ is the one-hot encoding of $y$, and $\sum_{i=1}^{M_1} + \sum_{j=1}^{M_2} = 1$. Together with the constraints of $\|x' - x\|_\infty \leq \epsilon$, their adversarial examples updating process can be described as follows:

$$x'_{t+1} = \text{Clip}_{x,\epsilon} \left\{ x'_t + \alpha \cdot \text{sign}\left(\nabla_x \mathcal{L}\left(x'_t, y\right)\right) \right\}, \tag{35}$$

$$\mathcal{L}\left(x'_t, y\right) = \sum_{i=1}^{M_1} w_i \mathcal{L}\left(h_{robust}^i\left(x'_t\right), y\right) + \sum_{j=1}^{M_2} w_j \mathcal{L}\left(h_{weak}^j\left(x'_t\right), y\right), \tag{36}$$

where the $h_{robust}^i$ and $h_{weak}^j$ represent robust models and weakly robust models respectively, $w$ is the coresponding ensemble weights.

**Adaptive Model Ensemble Attack (AdaEA).** Existing ensemble attacks simply fuse the outputs of agent models uniformly, and thus do not effectively capture and amplify the intrinsic transfer information of the adversarial examples. Chen et al. (2023a) propose AdaEA that adaptively controls the fusion of each model's output, via monitoring the discrepancy ratio of their contributions towards the adversarial objective. Then, an extra disparity-reduced filter is introduced to further synchronize the update direction. The basic idea of an ensemble attack is to utilize the output of multiple white box models to obtain an average model loss and then apply a gradient-based attack to generate adversarial examples. In their work, they propose AdaEA equipped with adaptive gradient modulation (AGM) and a disparity-reduced filter (DRF) to amend the gradient optimization process for boosting the transferable information in the generated adversarial examples. In detail, The AGM strategy can adaptively combine the outputs of each model through an adversarial ratio, thus increasing the strength of the transferable information in the generated adversarial examples. The DRF can decrease the differences between the agent models by calculating a discrepancy map and synchronizing the update direction. The process of AdaEA can be represented in short by the following equation:

$$x_{t+1}^{adv} = \text{Clip}_x^\epsilon \left\{ x_t^{adv} + \alpha \cdot \text{sign}\left(g_{t+1}^{ens}\right) \right\}, \tag{37}$$

$$g_{t+1}^{ens} = \nabla_{x_t^{adv}} \mathcal{L}\left(\sum_{k=1}^{K} w_k^* f_k\left(x_t^{adv}\right), y\right) \otimes \boldsymbol{B}, \tag{38}$$

where the $g$ represents the ensemble gradient of K models, the $\boldsymbol{B}$ represents a filter that can clean the disparity part in the ensemble gradient, and $\otimes$ denotes the element-wise multiplication.

## 3.4 Model Component-Based Transferability Enhancing Methods

In this subsection, we introduce the transferability-enhancing approaches based on various model components. Typically, the components are from the surrogate model in Equation 6.

**Features.** Several methods have aimed to improve transfer attacks, which focus on considering the feature space of the source model to generate noise that less overfits the specific architecture. Zhou et al. (2018) first demonstrated that maximizing the distance of intermediate feature maps between clean images and adversarial examples can enhance the transfer attack across models. They introduced two additional penalty terms in the loss function to efficiently guide the search directions for traditional untargeted attacks. The optimization problem can be described as follows:

$$x^{adv} = \arg\max_{\|x - x_{adv}\|^p \leq \epsilon} l(x_{adv}, t) + \lambda \sum_{d \in D} \left\| T(L(x, d) - T(L(x^{adv}, d)) \right\|^2 + \eta \sum_i \text{abs} R_i(x^{adv} - x, w_s), \tag{39}$$

where the first term represents the traditional untargeted attack loss, and $L(x, d)$ denotes the intermediate feature map in layer $d \in D$. Here, $T(L(x, d))$ signifies the power normalization (Perronnin et al., 2010) of $L(x, d)$. The regularization serves as a form of low-pass filter, enforcing the continuity of neighboring pixels and reducing the variations of adversarial perturbations. Naseer et al. (2018) and Hashemi et al. (2022) followed this idea and generated adversarial examples that exhibited transferability across different network architectures and various vision tasks, including image segmentation, classification, and object detection.

Huang et al. (2019a) proposed the Intermediate Level Attack (ILA), which involves fine-tuning an existing adversarial example by magnifying the impact of the perturbation on a pre-specified layer of the source model. Given an adversarial example $x'$ generated by a baseline attack, it serves as a hint. ILA aims to find a $x''$ such that the optimized direction matches the direction of $\Delta y_l' = F_l(x') - F_l(x)$ while maximizing the norm of the disturbance in this direction $\Delta y_l'' = F_l(x'') - F_l(x)$. Within this framework, they propose two variants, ILAP and ILAF. The ILAP simply adopts the dot product for the maximization problem, and the ILAF augments the losses by separating out the maintenance of the adversarial direction from the magnitude and controls the trade-off with the additional parameter $\alpha$.

$$\mathcal{L}_{ILAP}\left(y_l', y_l''\right) = -\Delta y_l' \cdot \Delta y_l'' \tag{40}$$

$$\mathcal{L}_{ILAF}\left(y_l', y_l''\right) = -\alpha \cdot \frac{\|\Delta y_l''\|_2}{\|\Delta y_l'\|_2} - \frac{\Delta y_l''}{\|\Delta y_l''\|_2} \cdot \frac{\Delta y_l'}{\|\Delta y_l'\|_2} \tag{41}$$

Salzmann et al. (2021) introduced a transferable adversarial perturbation generator that employs a feature separation loss, with the objective of maximizing the $L_2$ distance between the normal feature map $f_l(x_i)$ and the adversarial feature map $f_l(x_i^{adv})$ at layer $l$. This is defined as:

$$\mathcal{L}_{feat}(x_i, x_i^{adv}) = \left\| f_l(x_i) - f_l(x_i^{adv}) \right\|_F^2. \tag{42}$$

The above methods usually trap into a local optimum and tend to overfit to the source model by indiscriminately distorting features, without considering the intrinsic characteristics of the images. To overcome this limitation, Ganeshan et al. (2019) proposed the Feature Disruptive Attack (FDA), which disrupts features at every layer of the network and causes deep features to be highly corrupt. For a given $i^{th}$ layer $l_i$, they increase the layer objective $\mathcal{L}$:

$$\mathcal{L}\left(l_i\right) = D\left(\left\{l_i(\tilde{x})_{N_j} \mid N_j \notin S_i\right\}\right) - D\left(\left\{l_i(\tilde{x})_{N_j} \mid N_j \in S_i\right\}\right), \tag{43}$$

where $l_i(\tilde{x})_{N_j}$ denotes the $N_j$th value of $l_i(\tilde{x})$, $S_i$ denotes the set of activations that contribute to the current prediction. While this set is not straightforward to find, it can be approximated using a measure of central tendency, such as the median or the inter-quartile-mean. $D$ is a monotonically increasing function of activations $l_i(\tilde{x})$. They perform it at each non-linearity in the network and combine the per-layer objectives for the goal. FDA treats all neurons as important neurons by differentiating the polarity of neuron importance by mean activation values.

In contrast, Wang et al. (2021c) proposed a Feature Importance-aware Attack (FIA) to improve the transferability of adversarial examples by disrupting the critical object-aware features that dominate the decision of different models. FIA leverages feature importance, obtained by averaging the gradients with respect to feature maps from the source model, to guide the search for adversarial examples.

Zhang et al. (2022c) introduced the Neuron Attribution-based Attack (NAA), which is a feature-level attack based on more accurate neuron importance estimations. NAA attributes the model's output to each neuron and devises an approximation scheme for neuron attribution, significantly reducing the computation cost. Subsequently, NAA minimizes the weighted combination of positive and negative neuron attribution values to generate transferable adversarial examples.

Wu et al. (2020b) proposed to alleviate overfitting through model attention. They consider an entire feature map as a fundamental feature detector and approximate the importance of feature map $A_k^c$ (the $c$-th feature map in layer $k$) to class $t$ with spatially pooled gradients:

$$\alpha_k^c[t] = \frac{1}{Z} \sum_m \sum_n \frac{\partial f(\mathbf{x})[t]}{\partial A_k^c[m, n]}. \tag{44}$$

They scale different feature maps with corresponding model attention weights $\alpha_k^c[t]$ and perform channel-wise summation of all feature maps in the same layer. Then derive the attention map for the label prediction $t$ as follows:

$$H_k^t = \text{ReLU}\left(\sum \alpha_k^c[t] \cdot A_k^c\right), \tag{45}$$

Finally, they combine the original goal which aims to mislead the final decision of the target model, and the attention goal which aims to destroy the vital intermediate features.

$$\arg\max_\delta \quad \mathcal{L}\left(f\left(\mathbf{x}^{adv}\right), t\right) + \lambda \sum_k \left\|H_k^t\left(\mathbf{x}^{adv}\right) - H_k^t(\mathbf{x})\right\|^2. \tag{46}$$

The above transfer attack methods are only for un-targeted attacks. Rozsa et al. (2017) and Inkawhich et al. (2019) first describe a transfer-based targeted adversarial attack that manipulates feature space representations to resemble those of a target image. The Activation Attack (AA) loss is defined to make the source image $I_s$ closer to an image of the target class $I_t$ in feature space.

$$J_{AA}\left(I_t, I_s\right) = \left\|f_L\left(I_t\right) - f_L\left(I_s\right)\right\|_2 = \left\|A_t^L - A_s^L\right\|_2, \tag{47}$$

where $J_{AA}$ is the Euclidean distance between the vectorized source image activations and vectorized target image activations at layer L, and $f_L$ be a truncated version of a white-box model $f_w$. However, this method is challenging to scale to larger models and datasets due to the lack of modeling for the target class and its sensitivity to the chosen target sample.

Inkawhich et al. (2020b;a) propose to model the class-wise feature distributions at multiple layers of the white-box model, aiming for a more comprehensive representation of the target class in targeted attacks. Initially, they modeled the feature distribution of a DNN using an auxiliary Neural Network $g_{l,c}$ to learn $p(y = c|f_l(x))$, which represents the probability that the features of layer $l$ of the white-box model, extracted from input image $x$, belong to class $c$. Subsequently, the attack employed these learned distributions to generate targeted adversarial examples by maximizing the probability that the adversarial example originates from a specific class's feature distribution. Additionally, they developed a flexible framework that could extend from a single layer to multiple layers, emphasizing the explainability and interpretability of the attacking process.

Some other works have also explored the properties of adversarial examples in the feature space. Wang et al. (2021b) discovered a negative correlation between transferability and perturbation interaction units and provided a new perspective to understand the transferability of adversarial perturbations. They demonstrated that multi-step attacks tend to generate adversarial perturbations with significant interactions, while classical methods of enhancing transferability essentially decrease interactions between perturbation units. Therefore, they proposed a new loss to directly penalize interactions between perturbation units during an attack, significantly improving the transferability of previous methods.

Waseda et al. (2023) demonstrated that adversarial examples tend to cause the same mistakes for non-robust features. Different mistakes could also occur between similar models regardless of the perturbation size. Both different and the same mistakes can be explained by non-robust features, providing a novel insight into developing transfer adversarial examples based on non-robust features.

**Batch Normalization (BN).** Benz et al. (2021) investigate the effect of BN on deep neural networks from a non-robust feature perspective. Their empirical findings suggest that BN causes the model to rely more heavily on non-robust features and increases susceptibility to adversarial attacks. Further, they demonstrate that a substitute model trained without BN outperforms its BN-equipped counterpart and that early-stopping the training of the substitute model can also boost transferable attacks.

**Skip Connections.** Wu et al. (2020a) find that skip connections facilitate the generation of highly transferable adversarial examples. Thus, they introduced the Skip Gradient Method (SGM), which involves using more gradients from skip connections rather than residual ones by applying a decay factor on gradients. Combined with existing techniques, SGM can drastically boost state-of-the-art transferability.

**ReLU activation** Guo et al. (2020) propose to boost transferability by enhancing the linearity of deep neural networks in an appropriate manner. To achieve this goal, they propose a simple yet very effective method technique dubbed linear backpropagation (LinBP), which performs backpropagation in a more linear

fashion using off-the-shelf attacks that exploit gradients. Specifically, LinBP computes the forward pass as normal but backpropagates the loss linearly as if there is no ReLU activation encountered.

**Patch Representation.** Naseer et al. (2022) propose the Self-Ensemble (SE) method to find multiple discriminative pathways by dissecting a single ViT model into an ensemble of networks. They also introduce a Token Refinement (TR) module to refine the tokens and enhance the discriminative capacity at each block of ViT. While this method shows promising performance, it has limited applicability since many ViT models lack enough class tokens for building an ensemble, and TR requires is time-consuming. Wei et al. (2018) find that ignoring the gradients of attention units and perturbing only a subset of the patches at each iteration can prevent overfitting and create diverse input patterns, thereby increasing transferability. They propose a dual attack framework consisting of a Pay No Attention attack and a PatchOut attack to improve the transferability of adversarial samples across different ViTs.

**Ensemble of Models.** Liu et al. (2017) first propose transferable generating adversarial examples by utilizing an ensemble of multiple models with varying architectures. Gubri et al. (2022b) presents a geometric approach to enhance the transferability of black-box adversarial attacks by introducing Large Geometric Vicinity (LGV). LGV constructs a surrogate model by collecting weights along the SGD trajectory with high constant learning rates, starting from a conventionally trained deep neural network. Gubri et al. (2022a) develop a highly efficient method for constructing a surrogate based on state-of-the-art Bayesian Deep Learning techniques. Their approach involves approximating the posterior distribution of neural network weights, which represents the belief about the value of each parameter. Similarly, Li et al. (2023) adopt a Bayesian formulation in their method to develop a principled strategy for possible fine-tuning, which can be combined with various off-the-shelf Gaussian posterior approximations over deep neural network parameters. Huang et al. (2023) focus on the single-model transfer-based black-box attack in object detection. They propose an enhanced attack framework by making slight adjustments to its training strategies and draw an analogy between patch optimization and regular model optimization. In addition, they propose a series of self-ensemble approaches on the input data, the attacked model, and the adversarial patch to efficiently utilize the limited information and prevent the patch from overfitting.

## 4 Generation-Based Transferable Attacks

Optimization-based adversarial attacks discussed in the previous use gradients from surrogate models to iteratively optimize bounded perturbations for each clean image at test time. In this section, we introduce generation-based transferability-enhancing methods. This class of methods takes an alternative approach by directly synthesizing the adversarial example (or the adversarial perturbation) with generative models. Generation-based attacks comprise two stages: Training and the attack stages. During the training stage, the attacker trains a generative model $\mathcal{G}_\theta(\cdot)$, a function parameterized by $\theta$ that outputs either the adversarial example $x^{adv}$ or an adversarial perturbation $\delta$. The optimization of the generator parameters can be formulated as follows:

$$\max_\theta \mathbb{E}_{(x,y)} l(f_s(\mathcal{G}_\theta(\cdot)), y) \tag{48}$$

where $f_s(\cdot)$ is a surrogate model, and $\mathcal{G}_\theta(\cdot)$ directly generates the adversarial example. If the generator predicts the perturbation $\delta$ instead, the loss becomes $l(f_s(\mathcal{G}_\theta(\cdot) + x), y)$. In the case of targeted attacks, the optimization is described as:

$$\min_\theta \mathbb{E}_{(x,y_t)} l(f_s(\mathcal{G}_\theta(\cdot)), y_t) \tag{49}$$

Note that a surrogate model $f_s(.)$ is involved in the first step of the generative model-based attack. During the attack stage, adversarial examples are obtained directly with a single forward inference of the learned generator $\mathcal{G}_\theta(\cdot)$.

The input to the generator varies depending on the problem formulation. For input-dependent (Poursaeed et al., 2018; Naseer et al., 2019) adversarial perturbation generation, where the goal is to generate a perturbation specific to the given input $x$, we have:

$$x^{adv} = \mathcal{G}_\theta(x) \tag{50}$$

where any smoothing operations or additive and clipping operations can be absorbed into the mapping $\mathcal{G}$ (Alternatively, we can generate the perturbation instead of the $x^{adv}$, that is $\delta = \mathcal{G}_\theta(x)$). For universal adversarial perturbations (Poursaeed et al., 2018), where the perturbation can be added to any $x$, we input a fixed noise $z$ to the generator function:

$$\delta = \mathcal{G}_\theta(z) \tag{51}$$

Generative models are believed to possess several properties that can help achieve improved imperceptibility and transferability. Firstly, only one single model forward pass is required at test time once the generator is trained. This avoids the costly iterative perturbation process and thus, allows highly efficient adversarial attacks to be performed in an online fashion. Secondly, generators are less reliant on class-boundary information from the surrogate classifier since they can model latent distributions (Naseer et al., 2021). Finally, generative models provide latent spaces from which perturbations can be injected. This enables the search for adversaries in the lower-dimensional latent space rather than directly within the input data space, resulting in smoother perturbations with improved photorealism, diversity, and imperceptibility.

## 4.1 Methods Based on Unconditional Generation

**Generative Adversarial Pertruabtions (GAP)** Poursaeed et al. (2018) introduce generative models to the task of adversarial sample synthesis. In this work, the generator generates the perturbation ($\delta = \mathcal{G}_\theta(.)$) as opposed to generating the $x^{adv}$. They investigate two different variations: generating input-dependent adversarial perturbations and universal adversarial perturbations. For the former, they use

$$\max_\theta \mathbb{E}_{(x,y)} l(f_s(\mathcal{G}_\theta(x) + x), y) \tag{52}$$

as generator training objective where $l$ is defined as the cross entropy loss. They also investigate generating universal perturbations, where the perturbation $\delta$ can be added to any input image. In this case, the generator is given a fixed noise $z$ as input (51). Thus, the optimization term becomes:

$$\max_\theta \mathbb{E}_{(x,y)} l(f_s(\mathcal{G}_\theta(z) + x), y) \tag{53}$$

Their work demonstrates the high efficiency of learned generators for creating both targeted and untargeted adversarial examples.

**AdvGAN.** Xiao et al. (2018) proposed to incorporate adversarial training for the generator by introducing a discriminator $\mathcal{D}_\phi$ and solving a min-max game (equation 54). In addition to equation 48, they incorporated a GAN loss to promote the realism of synthesized samples and a soft hinge loss on the L2 norm, where $c$ denotes a user-specified perturbation budget.

$$\min_\phi \max_\theta \mathbb{E}_{(x)} \left( l(f_s(\mathcal{G}_\theta(x)), y) + \log(1 - \mathcal{D}_\phi(x)) + \log(\mathcal{D}_\phi(\mathcal{G}_\theta(x))) - \max(0, ||\mathcal{G}(x)||_2 - c) \right) \tag{54}$$

**Cross Domain Adversarial Perturbation.** Naseer et al. (2019) investigate the usage of generative models in generating adversarial attacks that transfer across different input domains. They propose to use relativistic loss in the generator training objective to enhance the transferability of cross-domain targeted attacks. The relativistic cross entropy (equation 55) objective is believed to provide a "contrastive" supervision signal that is agnostic to the underlying data distribution and hence achieves superior cross-domain transferability.

$$\mathcal{L} := \text{CE}(f_s(x) - f_s(\mathcal{G}_\theta(x))) \tag{55}$$

**Distribution and Neighbourhood Similarity Matching.** To achieve good transferability for cross-domain targeted attacks, Naseer et al. (2021) propose a novel objective that considers both global distribution matching as well as sample local neighborhood structures. In addition to solving the optimization problem in equation 48, they propose to add two loss terms (1) one that minimizes the (scaled) Jensen-Shannon Divergence between the distribution of perturbed adversarial samples from the source domain $p^s(\mathcal{G}_\theta(x))$ and the distribution of real samples from the target class in the target domain $p^t(x|y_t)$; (2) a second term

that aligns source and target similarity matrices $\mathbf{S}^s$ and $\mathbf{S}^t$ defined as $\mathcal{S}^s_{i,j} := \frac{f(x^i_s) \cdot f(x^j_s)}{\|f(x^i_s)\|\|f(x^j_s)\|}$ and $\mathcal{S}^t_{i,j} := \frac{f(x^i_t) \cdot f(x_s tj)}{\|f(x^i_t)\|\|f(x^j_t)\|}$, which serves the purpose of matching the local structures based on neighborhood connectivity.

$$\mathcal{L}_{aug} = D_{KL}(p^s(\mathcal{G}_\theta(x))\|p_t(x|y_t)) + D_{KL}(p^t(x|y_t)\|p^s(\mathcal{G}_\theta(x))) \tag{56}$$

$$\mathcal{L}_{sim} = D_{KL}(\mathbf{S}^t\|\mathbf{S}^s) + D_{KL}(\mathbf{S}^x\|\mathbf{S}^t) \tag{57}$$

**Attentive-Diversity Attack (ADA).** Kim et al. (2022) propose a method that stochasticly perturbs various salient features to enhance adversarial sample transferability. By manipulating image attention, their method is able to disrupt common features shared by different models and hence achieve better transferability. Their generator takes an image along with a random latent code $z \sim \mathcal{N}$ as input $\mathcal{G}_\theta(x, z)$. They propose two losses in addition to the classification loss in equation 48 : (1) $\mathcal{L}_{attn}$ that maximizes the distance between the attention maps of the original and the adversarial images for class-specific feature disruption and (2) $\mathcal{L}_{div}$ that promotes samples diversity by encouraging the generator to exploit the information in the latent code. They also argue that the stochasticity in $z$ can help circumvent poor local optima and extend the search space for adversarial samples.

$$\mathcal{L}_{attn} = \|A(\mathcal{G}_\theta(x, z)) - A(x)\|_2 \tag{58}$$

$$\mathcal{L}_{div} = \frac{\|A(\mathcal{G}_\theta(x_1, z_1)) - A(\mathcal{G}_\theta(x_2, z_2))\|_2}{\|z_1 - z_2\|} \tag{59}$$

**Conditional Adversarial Distribution (CAD)** Feng et al. (2022) propose a transferability-enhancing approach that emphasizes robustness against surrogate biases. They propose to transfer a subset of parameters based on CAD (i.e., the distribution of adversarial perturbations conditioned on benign examples) of surrogate models and learn the remainder of parameters based on queries to the target model while dynamically adjusting the CAD of the target model on new benign samples.

**Model Discrepancy Minimisation.** Zhao et al. (2023) propose an objective based on the hypothesis discrepancy principle that can be used to synthesize robust and transferable targeted adversarial examples with multiple surrogate models. In an adversarial training fashion, they jointly optimize the generator and the surrogate models (used as discriminators) to minimize the maximum model discrepancy (M3D) between surrogate models (equation 60), transform the image into a target class (equation 61) while maintaining the quality of surrogate models to provide accurate classification results on the original images (equation 62).

$$\max_{f_1, f_2} \min_\theta \mathcal{L}_d = \mathbb{E}_{x \sim \mathcal{X}} d[f_1 \circ \mathcal{G}_\theta(x), f_2 \circ \mathcal{G}_\theta(x)] \tag{60}$$

$$\min_\theta \mathcal{L}_a = \mathbb{E}_{x \sim \mathcal{X}} \text{CE}[f_1 \circ \mathcal{G}_\theta(x), y_t] + \text{CE}[f_2 \circ \mathcal{G}_\theta(x), y_t] \tag{61}$$

$$\max_{f_1, f_2} \mathcal{L}_c = \mathbb{E}_{x, y \sim (\mathcal{X}, \mathcal{Y})} \text{CE}[f_1(x), y] + \text{CE}[f_2(x), y] \tag{62}$$

## 4.2 Methods Based on Class-Conditional Generation

Early generative targeted attack methods (Poursaeed et al., 2018; Naseer et al., 2021) suffer from low parameter efficiency as they require training a separate generator for each class. To address this issue, various approaches have been proposed to construct conditional generative models that handle targeted attacks of different classes with a single unified model. While many different actualizations exist, these methods share the same mathematical formulation:

$$\min_\theta \mathbb{E}_{(x,y)} l(f_s(\mathcal{G}_\theta(x, y_t)), y_t) \tag{63}$$

**Conditional Generators.** Yang et al. (2022) propose a Conditional Generative model for a targeted attack, which can craft a strong Semantic Pattern (CG-SP). Concretely, the target class information was processed through a network before being taken as the condition of the generator (Mirza & Osindero, 2014). Claiming

that it is difficult for a single generator to learn distributions of all target classes, C-GSP divided all classes into a feasible number of subsets. Namely, only one generator is used for a subset of classes instead of each.

Various ways to inject the condition into the synthesis process have been explored. For example, some authors propose to add trainable embeddings (Han et al., 2019) that can add target class information to the input tensor. In a similar spirit, GAP++(Mao et al., 2020) extends GAP by taking target class encodings as model input and thereby only requires one model for both targeted and untargeted attacks. Multi-target Adversarial Network (MAN) (Han et al., 2019) propose a method that enables multi-target adversarial attacks with a single model by incorporating category information into the intermediate features. To further improve the adversarial transferability, Phan et al. (2020) propose a Content-Aware adversarial attack Generator (CAG) to integrate class activation maps (CAMs) (Zhou et al., 2016) information into the input, making adversarial noise more focused on objects.

**Diffusion Models.** Recent works have started to investigate the usage of diffusion models for enhancing adversarial transferability. DiffAttack (Chen et al., 2023b) is the first adversarial attack based on diffusion models (Ho et al., 2020), whose properties can help achieve imperceptibility. Concretely, the perturbations were optimized in the latent space after encoder and DDIM (Ho et al., 2020). Cross-attention maps are utilized in the loss function to distract attention from the labelled objects and disrupt the semantic relationship. Besides, self-attention maps are used for imperceptibility, keeping the original structure of images. In a similar spirit, AdvDiffuser (Chen et al., 2023d) and Adversarial Content Attack (ACA) (Chen et al., 2024) also leverage pre-trained diffusion models to craft highly transferable unrestricted adversarial examples.

# 5 Adversary Transferability Beyond Image Classification

In this section, we present transfer attacks beyond image classification tasks, such as various vision tasks and NLP tasks. Furthermore, the transferability across tasks is also summarized.

## 5.1 Transfer Attacks in Vision Tasks

**Image Retrieval.** Previous works (Yang et al., 2018; Tolias et al., 2019) have shown that image retrieval is also vulnerable to adversarial examples. Xiao & Wang (2021) explore the transferability of adversarial examples in image retrieval. In detail, they establish a relationship between the transferability of adversarial examples and the adversarial subspace by using random noise as a proxy. Then, they propose an adversarial attack method to generate highly transferable adversarial examples by being both adversarial and robust to noise. Xiao & Wang (2021) point out the relationship between adversarial subspace and black-box transferability. They propose to use additive Gaussian noise to estimate the generated adversarial region, thereby identifying adversarial perturbations that are both transferable and robust to additive noise corruption.

**Object Detection.** Wei et al. (2018) find that existing image object detection attack methods suffer from weak transferability, *i.e.,* the generated adversarial examples usually have an attack success rate in attacking other detection methods. Then they propose a generative attack method to enhance the transferability of adversarial examples by using the feature maps extracted by a feature network. Specifically, they adopt the Generative Adversarial Network (GAN) framework, which is trained by the high-level class loss and low-level feature loss. Cai et al. (2022b) propose an object detection attack approach to generate context-aware attacks for object detectors. Specifically, they adopt the co-occurrence of objects, their relative locations, and sizes as context information to generate highly transferable adversarial examples. Moreover, Staff et al. (2021) explore the impact of transfer attacks on object detection. Specifically, they conduct objectness gradient attacks on the advanced object detector, *i.e.,* YOLO V3. Then, they find increasing attack strength can significantly enhance the transferability of adversarial examples. They also study the transferability when the datasets for the attacking and target models intersect. They find the size of the intersection has a direct relationship with the transfer attack performance. Additionally, Cai et al. (2022a) have indicated that existing adversarial attacks could not effectively attack the context-aware object detectors. To address that, They propose a zero-query context-aware attack method that can generate highly transferable adversarial scenes to fool context-aware object detectors effectively.

**Segmentation.** Gu et al. (2021b) explore the transferability of adversarial examples on image segmentation models. In detail, they investigate the overfitting phenomenon of adversarial examples on both classification and segmentation models and propose a simple yet effective attack method with input diversity to generate highly transferable adversarial examples for segmentation models. Hendrik Metzen et al. (2017) explore the transferability of adversarial examples to attack the model of semantic image segmentation by generating universal adversarial perturbations. Specifically, they propose a method to generate universal adversarial examples that can change the semantic segmentation of images in arbitrary ways. The proposed adversarial perturbations are optimized on the whole training set.

**3D Tasks.** Previous works (Xiang et al., 2019; Zhou et al., 2019; Tsai et al., 2020) have developed several adversarial attack methods for 3D point clouds. Hamdi et al. (2020) discover that existing adversarial attacks of 3D point clouds lack transferability across different networks. Then, they propose an effective 3D point cloud adversarial attack method that takes advantage of the input data distribution by including an adversarial loss in the objective following Auto-Encoder reconstruction. Pestana et al. (2022) study the transferability of 3D adversarial examples generated by 3D adversarial textures and propose to use end-to-end optimization for the generation of adversarial textures for 3D models. Specifically, they adopt neural rendering to generate the adversarial texture and ensemble non-robust and robust models to improve the transferability of adversarial examples.

**Person Re-Identification.** Previous works (Gou et al., 2016; Xue et al., 2018; Huang et al., 2019b) have indicated that person re-identification (ReID), which inherits the vulnerability of deep neural networks (DNNs), can be fooled by adversarial examples.Wang et al. (2020) explore the transferability of adversarial examples on ReID systems. Specifically, they propose a learning-to-mis-rank method to generate adversarial examples. They also adopt a multi-stage network to improve the transferability of adversarial examples by extracting transferable features.

**Face Recognition.** Jia et al. (2022a) have indicated that previous face recognition adversarial attack methods rely on generating adversarial examples on pixels, which limits the transferability of adversarial examples. Then, they propose a unified, flexible adversarial attack method, which generates adversarial for perturbations of different attributes based on target-specific face recognition features to boost the attack transferability.

**Video Classification.** Wei et al. (2022a) have found that existing video attack methods have only limited transferability. Then, they propose a transferable adversarial attack method based on the temporal translation of the video, which generates adversarial perturbations over temporally translated video clips to enhance the attack transferability.

## 5.2 Transfer Attacks in NLP Tasks.

Yuan et al. (2020) introduce a comprehensive investigation into the transferability of adversarial examples for text classification models. In detail, they thoroughly study the impact of different factors, such as network architecture, on the transferability of adversarial examples. Moreover, they propose to adopt a generic algorithm to discover an ensemble of models capable of generating adversarial examples with high transferability. Furthermore, He et al. (2021) demonstrate the ability of an adversary to compromise a BERT-based API service. With the available model, they can generate highly transferable adversarial examples. Wang et al. (2022) show that the adversarial examples are also transferable across the topic models, which are important statistical models. To further improve the transferability, they propose to use a generator to generate effective adversarial examples and an ensemble method, which finds the optimal model ensemble.

With the rise of large language models (LLM) like BERT (Kenton & Toutanova, 2019), GPT (Brown et al., 2020b), and their variants (Li et al., 2019), the adversarial examples on them have also received attention. Recently, Zou et al. (2023) have demonstrated it is possible to induce aligned language models to generate inappropriate content, dubbed jailbreak attack. They also propose a simple way to make the jailbreak attack more transferable to other LLMs. Specifically, a jailbreak attack aims to maximize the likelihood of the language model generating an affirmative response instead of declining to answer. They implement the attack by identifying a suffix that, when appended to various queries given to a language model, encourages generating undesirable content. They improve the transferability by attacking multiple surrogate LLMs with

a single suffix. Zou et al. (2023) show that the adversarial suffix is even transferable to several mainstream close-sourced language models, e.g. ChatGPT (Achiam et al., 2023).

### 5.3 Cross-Task Transfer Attacks.

Naseer et al. (2018) propose a novel adversarial attack method, which adopts the neural representation distortion to generate adversarial examples. They have demonstrated the remarkable transferability of adversarial examples across different neural network architectures, datasets, and tasks. Naseer et al. (2019) propose a novel concept of domain-invariant adversaries, which demonstrates the existence of a shared adversarial space among different datasets and models. They introduce a new generative framework that creates strong adversarial examples with a relativistic discriminator, outperforming traditional instance-specific attacks with a universal adversarial function. Moreover, they propose to exploit the adversarial patterns capable of deceiving networks trained on completely different domains to improve attack transferability. Lu et al. (2020) investigate the transferability of adversarial examples across diverse computer vision tasks, which include object detection, image classification, semantic segmentation, etc. They propose a Dispersion Reduction (DR) adversarial attack method which minimizes the standard deviation of intermediate feature maps to disturb features that are used by models intended for various tasks. Wei et al. (2022b) study the transferability of adversarial perturbations across different modalities. In detail, they apply the adversarial examples on white-box image-based models to attacking black-box video-based models by exploiting the similarity of low-level feature spaces between images and video frames. Naseer et al. (2023) propose to adopt task-specific prompts to incorporate spatial (image) and temporal (video) cues into the same source model, which can enhance the transferability of attacks from image-to-video and image-to-image models. They propose a method to add dynamic cues to pre-trained image models through a simple video-based transformation. Lu et al. (2023) study the adversarial transferability of some vision-language pre-training models. They propose a set-level guidance adversarial attack to improve the transferability of adversarial examples on vision-language pre-training models, which makes full use of cross-modal guidance. Han et al. (2023a) propose adopting optimal transport optimization to enhance the adversarial transferability of vision-language models, which uses optima transmission theory to find the most effective mapping between image and text features. Hu et al. (2024) propose to attack intermediate features of an encoder pre-trained on vision-language data for cross-task transferability. They adopt a patch-wise approach that independently diverts the representation of each adversarial image patch from the corresponding clean one by minimising the cosine similarity between the two, thereby producing highly transferable adversaries that fool various vision-language understanding tasks.

## 6 Challenges, Opportunities and Connections to Broader Topics

This section delves into the intricacies of the challenges and illustrates the promising avenues for better transferability-based attacks, and their evaluation and understanding.

### 6.1 Challenges and Opportunities

**Adversarial Transferability is Far from Perfect.** Adversarial examples are far from achieving perfection when transferred to other models due to several inherent challenges. First, the performance of adversarial transferability tends to degrade when evaluated on a variety of neural network architectures, highlighting the inconsistency in transferability across different models (Yu et al., 2023). Secondly, the task of transferring the adversarial perturbations created by targeted adversarial attacks remains challenging. Misleading to a specific class is much more difficult than a simple fool in the case of adversarial transferability (Yu et al., 2023). Finally, the current transferability-enhancing methods are mainly developed to target visual classification models with predefined classes. The current vision-language models (Lu et al., 2023; Radford et al., 2021b; Alayrac et al., 2022; Chen et al., 2023c), which extract visual information from a distinct perspective, pose unique challenges for transferability. These issues indicate that more transferability-enhancing methods should be explored for better defense evaluation.

**Natural, Unrestricted and Non-Additive Attacks.** Albeit out of the scope of this survey, we note the alternative, relaxed definition of adversarial attacks does exist. Adversarial perturbation does not need to be constrained by any norm-ball Hendrycks et al. (2021); Zhao et al. (2017) and can be constructed through means other than additive noise (e.g. through transformations) (Brown et al., 2018). Several studies have explored the transferability of natural adversarial attacks Chen et al. (2023d), unconstrained (unrestricted) adversarial attacks (Liu et al., 2023; Chen et al., 2023b; Gao et al., 2022) and non-additive attacks (Zhang et al., 2020). Nonetheless, the community has not yet reached a consensus on how to effectively evaluate such attacks. For example, perceptual metrics other than $L_p$ distances may be required to evaluate stealthiness.

**Source Model for Better Transferability.** Current transferability methods are typically post hoc approaches that involve enhancing the ability of adversarial examples generated on pre-trained models to deceive other models. When considering the source model, the question arises: How can we train a model to improve the transferability of adversarial examples created on them? For instance, one promising avenue for achieving this is to learn the model from the perspective of knowledge transferability. A model with high knowledge transferability inherently becomes a superior source model, as adversarial examples generated on it exhibit a greater capacity to successfully deceive other models (Liang et al., 2021; Xu et al., 2022b). A follow-up question is which model architectures transfer better to others, CNNs, Vision Transformers (Naseer et al., 2022; Wu et al., 2021; Ma et al., 2023; Wu et al., 2022), Capsule Networks (Gu et al., 2021a), or Spiking Neural Networks (Xu et al., 2022a).

**Relation to Transferability Across Image Samples**. In this work, we focus on adversarial transferability across models, namely, the ability of an adversarial perturbation crafted for one model or set of data to successfully fool another model. The community has also found that an adversarial perturbation that effectively fools one image can also be applied to a different image to achieve a similar adversarial effect, which is referred to as adversarial transferability across image samples (i.e. Universal Adversarial Image) (Moosavi-Dezfooli et al., 2017). Understanding the interplay between transferability across images and transferability across models is essential for comprehending the broader landscape of adversarial attacks and defences. These two dimensions together define the versatility and robustness of adversarial perturbations.

**Theoretical Perspectives on Adversarial Transferability.** The root causes behind transferability receive continued research interest. Demontis et al. (2019) examine the effect of two factors on attack transferability: the intrinsic adversarial vulnerability of the target model and the complexity of the surrogate model. Ilyas et al. (2019) attributes adversarial transferability to the presence of non-robust features and points out the potential misalignment between robustness and inherent data geometry. Waseda et al. (2023) extends the theory of non-robust features by examining "class-aware transferability". In particular, they differentiate between the cases in which a target model predicts the same wrong class as the source model or a different wrong class, drawing connections between adversarial vulnerabilities and models' tendency of learning *superficial cues* (Jo & Bengio, 2017) and *shortcuts* (Geirhos et al., 2020). Charles et al. (2019) examine adversarial transferability from a geometric point of view and prove the existence of *transferable adversarial directions* for simple network architectures. Based on observations that AEs tend to occur in contiguous regions within which all points can similarly fool the model (referred to as *adversarial subspaces*) (Tanay & Griffin, 2016), Tramèr et al. (2017) explains the transferability as a result of intersection of models' adversarial subspaces: a higher number of orthogonal adversarial directions within these subspaces often implies higher transferability. Building on these works, Gubri et al. (2022b) highlights the role of weight space geometry in adversarial transferability, showing that adding random Gaussian noise to the weight space of DNNs increases their potential as surrogates for crafting more transferable adversaries. A contemporary work by Zhu et al. (2021) makes an analogous observation regarding the effect on adversarial transferability of adding random Gaussian noise in the output space. They propose Intrinsic Adversarial Attack (IAA) to diminish the impact of the deeper model layers. By doing so, they effectively exploit low-density regions of the data distribution where many highly transferable AEs can be found.

**Evaluation Metrics**. The assessment of adversarial transferability is a complex undertaking that demands a thorough and extensive set of metrics. The Fooling Rate as a popular choice is often used to quantify the transferability of adversarial examples. It gauges the effectiveness of adversarial perturbations by measuring the percentage of these perturbations that can successfully deceive a specified target model. However, it's important to emphasize that the Fooling Rate metric is highly contingent on the choice of target models,

which introduces a considerable source of variability into the evaluation process. Recent research, as highlighted in (Yu et al., 2023), has illuminated the profound impact that the selection of target models can have on the relative rankings of different transferability-enhancing methods. Consequently, there is a pressing need for even more comprehensive evaluation metrics.

Consequently, there is a pressing need for an even more comprehensive benchmark that can encompass a wider range of model architectures and configurations. In addition to empirical evaluations, there is also a growing recognition of the necessity for theoretical characterizations of transferability. Such theoretical analyses can provide valuable insights into the underlying principles governing the transferability of adversarial attacks.

**Benchmarking Adversarial Transferability.** Various benchmarks have been developed to evaluate adversarial transferability. Initial robustness benchmarks include transfer-based black-box attacks to evaluate the adversarial robustness of models (Croce et al., 2020; Dong et al., 2020). (Zhao et al., 2022) evaluates the transferability of adversarial examples, considering the perspectives of stealthiness of adversarial perturbations. (Mao et al., 2022) evaluates transferability-based attacks in real-world environments. Furthermore, (Zhao et al., 2022) builds a more reliable evaluation benchmark by including various architectures.

**Hybrid Approaches Combining Optimization-based and Generation-based.** Both optimization-based and generation-based approaches have been intensively studied. While each of them has both advantages and limitations, the pursuit of a well-designed hybrid method that combines both approaches to achieve better transferability is a promising direction for future endeavours. For example, certain methods leverage generative models while also going through optimization iterations during inference (Chen et al., 2023d). Such hybrid methods have the potential to leverage the strengths of each approach to enhance the robustness and generalization of adversarial examples across various models and scenarios.

**Adversarial Transferability Across Large Multimodal Models.** The transferability of adversarial examples across language models has been studied by various works introduced in section 5.2. Given the prevalence of large language models (LLM) and multimodal foundation models, the adversarial transferability across such foundation models also become increasingly relevant to the community. The pioneering work of Zhao et al. (2024) shows that adversarial examples crafted against pre-trained models such as CLIP (Radford et al., 2021a) and BLIP (Li et al., 2022) can be transferred to other multimodal foundation models such as MiniGPT-4 (Zhu et al., 2023a) and LLaVA (Liu et al., 2024). Similarly, Dong et al. (2023) demonstrates the feasibility of attacking Google's Bard with vision encoders of open-sourced models. Meanwhile, various works demonstrate that adversarial examples created on CLIP can be transferred to various CLIP-based systems (Lu et al., 2023; Zhang et al., 2022b; Han et al., 2023a; Hu et al., 2024). As reported in the current work (Zhao et al., 2024; Dong et al., 2023; Luo et al., 2024), the transferability across large multimodal models is still very limited. Exploring the root causes behind such limited transferability and identifying strategies for enhancing it could be an interesting direction for future research.

## 6.2 Connections to Broader Topics

**Relation to Adversarial Transferability Prior To Deep Learning Era** The concept of adversarial examples holds relevance both in deep learning contexts and in earlier eras of machine learning. Prior research shows that traditional machine learning algorithms also suffer from adversarial examples, e.g., support vector machine (Papernot et al., 2016) and decision tree (Papernot et al., 2016; Biggio et al., 2013). Furthermore, Papernot et al. (2016) show that the adversarial examples can be transferred to different classes of machine learning classifiers. Specifically, the adversarial examples created on traditional machine learning models can be transferred to others, even deep neural network-based classifiers. Similarly, the ones created on deep neural networks can also be transferred to traditional machine learning classifiers. However, the transferability is only shown on toy datasets. The experiments on large-scale datasets (e.g. ImageNet-1k (Russakovsky et al., 2015)) are infeasible since the traditional algorithms are not scalable to large datasets.

**Relation to Trustworthy AI.** Adversarial transferability is closely linked to Trustworthy AI, which aims to make AI systems reliable, fair, transparent, and accountable (Kaur et al., 2022). When adversarial examples fool one AI model and then trick other models too, it highlights how vulnerable AI systems can be. Studying transferability can help us understand the cause of the adversarial example, namely, a better understanding

of failure cases of AI systems (Ilyas et al., 2019; Waseda et al., 2023). In addition, understanding this link helps improve AI's reliability by developing better defenses against such tricks (Madry et al., 2017).

**Relation to Adversarial ML.** Adversarial ML focuses on understanding and mitigating vulnerabilities in machine learning models (Oprea & Vassilev, 2023). In adversarial ML, the goal is to investigate how malicious actors can manipulate or deceive ML systems by modifying inputs (i.e. adversarial example) to cause incorrect predictions or behaviors. Adversarial transferability is a crucial idea in Adversarial ML. It shows how adversarial examples, which are inputs crafted to fool one ML model, can also fool other models, even if they are different (Goodfellow et al., 2014; Papernot et al., 2016). This reveals how vulnerabilities in ML systems can spread widely. Understanding adversarial transferability helps researchers grasp the complexities of attacks on ML models and develop stronger defenses against them.

# 7    Conclusion

In this comprehensive survey, we embarked on a journey through the intricate world of adversarial transferability. Transferability allows adversarial examples designed for one model to successfully deceive a different model, often with a distinct architecture, opening the door to black-box attacks. The adversarial transferability across DNNs raises concerns about the reliability of DNN-based systems in safety-critical applications like medical image analysis and autonomous driving.

Throughout this survey, we navigated through the terminology, mathematical notations, and evaluation metrics crucial to understanding adversarial transferability. We explored a plethora of techniques designed to enhance transferability, categorizing them into two main groups: surrogate model-based and generative model-based methods. Moreover, we extended our investigation beyond image classification tasks, delving into transferability-enhancing techniques in various vision and natural language processing tasks, as well as those that transcend task boundaries.

As the DNN landscape continues to advance, the comprehension of adversarial examples and their transferability remains crucial. By illuminating the vulnerabilities inherent in these systems, we aim to contribute to the development of more resilient, secure, and trustworthy DNN models, ultimately paving the way for their safe deployment in real-world applications. In this ever-evolving journey towards adversarial resilience, we hope that this survey will serve as a valuable resource for researchers, practitioners, and enthusiasts alike.

**Broader Impact Statement.** At the intersection of machine learning and security, the study of adversarial examples and their transferability not only illuminates the vulnerabilities of modern learning systems but also opens avenues for strengthening their robustness and reliability. By comprehensively surveying the landscape of adversarial transferability, this paper contributes to a deeper understanding of the challenges posed by adversarial attacks across diverse domains, from image classification to various tasks. As researchers and practitioners strive to fortify machine learning models against adversarial manipulation, the insights gleaned from this survey serve as a compass, guiding the development of more resilient algorithms and informing strategies for defending against emerging threats. Overall, this paper aims to better understand and safeguard against adversarial exploitation.

### Acknowledgments

This work is partially supported by the UKRI grant: Turing AI Fellowship EP/W002981/1 and EPSRC/MURI grant: EP/N019474/1. We would also like to thank the Royal Academy of Engineering.

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

## A More Variants of I-FGSM

In this appendix, we first recall some background information on I-FGSM and present more variants of I-FGSM Dong et al..

We focus on perturbations constrained by an $\ell_\infty$ ball with radius $\epsilon$, that is, $||x^{adv} - x||_\infty \le \epsilon$. To understand the rest of this section, we begin by formalizing the iterative variant of the fast gradient sign method (I-FGSM) (Goodfellow et al., 2014), which serves as the basis for the development of other methods. The I-FGSM has the following update rule:

$$g^{(t+1)} = \nabla\ell(x^{adv(t)}, y),$$
$$x^{adv(t+1)} = \text{Clip}_x^\epsilon\{x^{adv(t)} + \alpha \cdot \text{sign}(g^{(t+1)})\}, \tag{64}$$

where $g^{(t)}$ is the gradient of the loss function with respect to the input, $\alpha$ denotes the step size at each iteration, and $\text{Clip}_x^\epsilon$ ensures that the perturbation satisfies the $\ell_\infty$-norm constraints. More variants of I-FGSM are as follows:

**Nesterov (NI-FGSM).** Nesterov Accelerated Gradient (NAG) is another popular extension of the vanilla gradient descent algorithm that incorporates momentum to accelerate convergence and improve the generalization of neural networks (Nesterov, 1983). On top of the momentum mechanism, the most distinct feature of NAG is that the gradient is evaluated at a lookahead position based on the momentum term. Lin et al. propose NI-FGSM (Nesterov Iterative Fast Gradient Sign Method), which integrates NAG in the iterative gradient-based attack to leverage its looking-ahead property to help escape from poor local optima. At each iteration, NI-FGSM first moves the data point based on the accumulated update $g^{(t)}$

$$x^{nes(t)} = x^{adv(t)} + \alpha \cdot \mu \cdot g^{(t)},$$

then we have

$$g^{(t+1)} = \mu \cdot g^{(t)} + \frac{\nabla\ell(x^{nes(t)}, y)}{||\nabla\ell(x^{nes(t)}, y)||_1},$$

and the formulation for $x^{adv(t+1)}$ remains the same as (17). The author argues that the anticipatory update of NAG helps to circumvent getting stuck at the local optimal easier and faster, thereby improving the transferability of the perturbation.

**Adam (AI-FGTM).** Adam is a popular adaptive gradient method that combines the first- and second-order momentum of the gradients (Kingma & Ba, 2015). Zou et al. introduce the Adam Iterative Fast Gradient Tanh Method (AI-FGTM), which adapts Adam to the process of generating adversarial examples. In addition to using Adam as opposed to the momentum formulation, a key feature of AI-FGTM is the replacement of the sign function with the tanh function, which has the advantage of a smaller perturbation size.

$$m^{(t+1)} = m^{(t)} + \mu_1 \cdot \nabla\ell(x^{adv(t)}, y),$$
$$v^{(t+1)} = v^{(t)} + \mu_2 \cdot (\nabla\ell(x^{adv(t)}, y))^2,$$

where $m^{(t)}$ denotes the first moment vector, $v^{(t)}$ represents the second moment vector, $\mu_1$ and $\mu_2$ are the first and second-order momentum factors, respectively. Instead of using a fixed step size of $\alpha$ in each iteration, AI-FGTM computes an adaptive step size based on

$$\alpha^{(t)} = \frac{\varepsilon}{\sum_{s=0}^t \frac{1-\beta_1^{(s+1)}}{\sqrt{\left(1-\beta_2^{(s+1)}\right)}}} \frac{1-\beta_1^{(t+1)}}{\sqrt{\left(1-\beta_2^{(t+1)}\right)}},$$

where $\beta_1$ and $\beta_2$ are exponential decay rates, $\lambda$ denotes a scale factor, and we have $\sum_{s=0}^{t-1} \alpha^{(s)} = \epsilon$. Finally, the update rule of AI-FGTM is

$$x^{adv(t+1)} = \text{Clip}_x^{\epsilon}\{x^{adv(t)} + \alpha^{(t)} \cdot \tanh(\lambda \frac{m^{(t+1)}}{\sqrt{v^{(t+1)}} + \delta})\},$$

where $\delta$ is to avoid division-by-zero.

**Variance Tuning (VNI/VMI-FGSM).** Previous work shows that the stochastic nature of the mini-batch gradient introduces a large variance in the gradient estimation, resulting in slow convergence and poor generalization; and this gives rise to various variance reduction methods (Roux et al., 2012; Johnson & Zhang, 2013). In the context of generating adversarial examples, Wang & He presents a variance-tuning technique that adopts the gradient information in the neighborhood of the previous data point to tune the gradient of the current data point at each iteration. Given an input $x \in \mathcal{R}^d$, they propose to approximate the variance of its gradient using

$$V(x) = \frac{1}{N}\sum_{i=1}^{N} \nabla_{x^i}\ell(x^i, y) - \nabla\ell(x, y), \tag{65}$$

where $x^i = x + r_i$ and each dimension of $r_i$ is independently sampled from a uniform distribution between $-\beta$ and $\beta$ with $\beta$ being a hyper parameter. They introduce Variance-tuning MI-FGSM (VMI-FGSM) and Variance-tuning NI-FGSM (VNI-FGSM) as an improvement over the original formulation. To integrate gradient variance in the iterative process, we can modify the following update rule for $g^{(t+1)}$ by using

$$\hat{g}^{(t+1)} = \nabla\ell(x^{(t)}, y)$$
$$g^{(t+1)} = \mu \cdot g^{(t)} + \frac{\hat{g}^{(t+1)} + v^{(t)}}{||\hat{g}^{(t+1)} + v^{(t)}||_1},$$
$$v^{(t+1)} = V(x^{(t)}) \tag{66}$$

where $x^{(t)} = x^{adv(t)}$ in MI-FGSM and $x^{(t)} = x^{nes(t)}$ in VNI-FGSM.

