# OpenReview forum: "A Survey on Transferability of Adversarial Examples Across Deep Neural Networks"
_TMLR — Accepted by TMLR_

### Review · Reviewer_ifNX · 2024-01-25

**Summary Of Contributions:**

The paper offers a comprehensive summary of prior research in the field of adversarial transferability. The author skillfully categorizes the existing approaches into two distinct methodologies: first, the surrogate model-based adversarial transferability, and second, the adversarial transferability utilizing generative models. Moreover, the author broadens the scope of discussion by introducing a variety of research contributions in natural language processing (NLP) and various vision tasks beyond the conventional realm of image classification. This inclusive approach enriches our understanding of the topic and its diverse applications.

**Audience:**

Yes

**Claims And Evidence:**

Yes

**Requested Changes:**

Major comments
* Offering insights or perspectives on the potential future directions of adversarial transferability research would add significant value to the paper.
Minor comments
* There appears to be an inconsistency in the use of `\citet` and `\citep`. For improved clarity, it's advisable to use `\citet` when the citation is a subject of the sentence and `\citep` in other cases. Consistent application of these citation formats would enhance the paper's readability and scholarly accuracy.
* In Section 3.1, there's a mismatch between the paragraph titles and their subjects. For example, DIM is mentioned as the subject in the first sentence, but SIM is used as the paragraph title. A similar issue is noticed with the usage of RDI, which might be intended as a subject but seems to have a grammatical error. Aligning the paragraph titles with the subjects discussed within each would improve the structure and coherence of this section.
* A typo error is present in the section titled "Challenges and Opportunities."
* The paper contains numerous grammatical errors. A comprehensive proofreading and editing process is recommended to enhance readability and professional presentation.

**Strengths And Weaknesses:**

**Strengths:**
* The survey paper is well-organized, effectively structuring the existing body of research on adversarial transferability.
* It includes an extensive survey of adversarial transferability, commendably encompassing both NLP tasks and a variety of vision tasks.
* The paper does an excellent job in clearly outlining the current challenges faced in the field of adversarial transferability.

**Weaknesses:**
* While the survey provides a comprehensive overview, it would be greatly enhanced by a more general conclusions and common findings. Offering insights or perspectives on the potential future directions of adversarial transferability research would add significant value to the paper.
* The paper, although informative, is somewhat marred by numerous grammatical errors. A thorough review and editing for language accuracy would enhance its clarity and professional impact.

---

> ### Author Response · Authors · 2024-03-20
> **Reponse to review comment**
>
> We thank the reviewer for the positive feedback on the well-organized paper structure, literature comprehensiveness, and clear outline of our survey.
>
> Our responses to the weaknesses and requested changes are as follows:
>
> **Q1**: While the survey provides a comprehensive overview, it would be greatly enhanced by a more general conclusions and common findings. Offering insights or perspectives on the potential future directions of adversarial transferability research would add significant value to the paper.
>
> **A1**: Thank you for the suggestions. Following the reviewer’s suggestion, we extend section 6 into multiple subsections. In section 6.1, we discuss the challenges we summarized from the current literature as well as potential future directions as opportunities.
>
> **Q2**: The paper, although informative, is somewhat marred by numerous grammatical errors. A thorough review and editing for language accuracy would enhance its clarity and professional impact.
>
> **A2**: Thank the reviewer for pointing this out. We proofread for multiple rounds with the help of native speakers and fixed some typos and grammatical errors (the fixed content is marked in blue).
>
> **Q3**: There appears to be an inconsistency in the use of \citet and \citep. For improved clarity, it's advisable to use \citet when the citation is a subject of the sentence and \citep in other cases.
>
> **A3**: We appreciate a lot for the reviewer’s suggestion regarding the format of the citation. Following the suggestion, we fix all the citations with the suggested format.
>
> **Q4**: In Section 3.1, there's a mismatch between the paragraph titles and their subjects. A similar issue is noticed with the usage of RDI, which might be intended as a subject but seems to have a grammatical error.
>
> **A4**: Thanks for the notice. We fixed them where it is possible. Please also note that some methods are inspired by previous ones. When introducing some methods, we might start with a previous method and introduce how they inspire the method of the subject of the paragraph. That is where the wrong impression about the mismatch comes from. We made them more clear now.
>
> **Q5**: A typo error is present in the section titled "Challenges and Opportunities." The paper contains numerous grammatical errors. A comprehensive proofreading and editing process is recommended to enhance readability and professional presentation.
>
> **A5**: Thanks for the suggestion. We did a comprehensive proofreading and editing process for multiple rounds with the help of native speakers. We will continue to improve the readability of our paper during our rebuttal time.
>
> We thank the reviewer again for their encouraging positive feedback and suggestions for improving our paper.

---

### Review · Reviewer_7rTq · 2024-02-13

**Summary Of Contributions:**

This paper summarizes the existing literature on adversarial example transferability. Specifically, the authors cover a massive amount of research on transferability, organized by various topics: how to assess the transferability, two categories (surrogate model-based and generative model-based) of transferability-enhancing methods, and transferability-enhancing methods in diverse domains. The authors also discuss the current challenges and research opportunities in adversarial transferability.

The paper introduces preliminary materials to cover the basic knowledge of adversarial attack transferability. Additionally, the authors summarize the three existing evaluation metrics that measure adversarial transferability: Fooling rate, interest class rank, and knowledge transfer-based metrics.

The next two sections (Section 3 and Section 4) are the main parts that cover transferability-enhancing methods, mainly on image classification tasks. Specifically, the authors divide the existing transferability-enhancing techniques into two categories: One is based on the surrogate model, and the other is based on the generative model. The surrogate model-based method uses a surrogate model that approximates the target model, and the attacks described in Section 3 aim to improve the transferability of adversarial examples when attacking the surrogate model. The authors introduce an optimization form of the surrogate model-based attack. Then, the authors further divide this category into four subcategories according to the part on which the methods concentrate: data augmentation, optimization, model, and loss. The generative model-based method uses a generative model to generate adversarial examples (or adversarial perturbations). The authors introduce generative model-based methods in Section 4 from two perspectives: The methods' effectiveness and efficiency.

Section 5 covers transferability-enhancing methods in machine learning domains other than image classification. Specifically, the authors briefly go through transfer attacks in vision tasks (e.g., object detection, segmentation, 3D point clouds, video, etc.), transfer attacks in NLP tasks, and even domain-invariant adversarial attacks.

Finally, the authors list some existing challenges in adversarial transferability research, e.g., better choices of surrogate model, how to evaluate or benchmark transferability, etc.

**Audience:**

Yes

**Broader Impact Concerns:**

I don’t see a particular broader impact concern regarding this paper.

**Claims And Evidence:**

Yes

**Requested Changes:**

1. Consider renaming the two categories of transferability-enhancing methods, i.e., surrogate model-based and generative model-based.
   * The current category names might look like generative model-based methods have nothing to do with the surrogate model, but they also use a surrogate model in the formula.
   * The generative model is a method that generates adversarial examples. However, the surrogate model is not an attack method, but the optimization generates adversarial examples for the methods in Section 3. I’d name this category optimization-based methods. (This may require renaming the subcategory in Section 3.2, though.)
2. I can see a few typos in mathematical notations/formulae. I’ll list some of those mistakes, but I cannot guarantee that I covered all the formulae. Please review all the formulae and ensure there are no typos.
   * In Table 2, $H_k^i$ should be $H_k^l$ because it is about $l$-th layer.
   * Equation 4 should be corrected to $\alpha_2^{f_s\rightarrow f_t}=\|\mathbb{E}_{x\sim D}[\widehat{\Delta _{f_s\rightarrow f_t}(x)} \cdot \widehat{\Delta _{f_s\rightarrow f_t}(x)}^\top]\|$. (i.e., move $\top$ inside the brackets.
   * In Equation 8, exchange the probability $p$ and $1-p$. I checked Xie et al. paper, and $p$ is the probability of applying the transformation.
   * In **Natural, Unrestricted and Non-Additive Attacks** (Page 21), $L-p$ in the last sentence looks like a typo of $L_p$
3. Sometimes, the authors use terms or notations without explaining what they mean. Please describe the missing parts.
   * In **ADMIX** description (Page 6), the authors used the term "Mixup" without explaining it. Please explain Mixup briefly with a citation.
   * In **AI-FGTM** description (Page 7), the adaptive step size uses an unseen notation $T$. I believe that the author wanted to write $\alpha^{(t)} = \frac{\varepsilon}{ \sum_{s=0}^t \frac{1-\beta_1^{s+1}}{\sqrt{1-\beta_2^{s+1}}}} \frac{1-\beta_1^{t+1}}{\sqrt{1-\beta_2^{t+1}}}$ (I changed the running index to be $s$, running from 0 to $t$)?
   * In Equation (31), the authors used an unseen notation $\gamma$ without explanation. I had to read Fang et al. to understand what it means. It looks like this $\gamma$ is some decay factor that determines the augmented model $f$, so Fang et al. replaced $f$ by $\gamma$. I think the authors tried to clarify this in the next line but made another mistake in the formula. Correct the formula to $G(\mathbf{x} _{adv}, \gamma)=G(\mathbf{x} _{adv}, f) = \nabla _{\mathbf{x} _{adv}} \mathcal{L}(f _{\mathbf{x} _{adv}}, y)$ and explain the context and clarify the notation.
   * In **Conditional Adversarial Distribution (CAD)** (Page 17), the authors mention that Feng et al. proposed to transfer a subset of parameters based on CAD, but the authors do not explain anything about CAD. I can only see that it is an abbreviation of Conditional Adversarial Distribution, but what does it even mean? Please describe the concept to make it more understandable.
4. In Table 1, the untargeted attack success is determined by the misclassification of a model. However, Equation 1 uses "change of prediction" as an attack success. Either change the description of the untargeted attack in Table 1 or change Equation 1 to $\arg\max_i f_t^i(x^{adv})\ne y,\text{ given }\arg\max_i f_s^i(x^{adv})\ne y$.
5. The first paragraph in Section 3.2 contains an interesting perspective. Can we interpret other methods (out of Section 3.2) from this viewpoint, i.e., data augmentation as another approach to enhance the generalization?
6. In Section 3.2, separate the iterative variants of FGSM from the methods for adversarial transferability. You can add a `\subsubsection` summarizing the variants or move them to other sections, e.g., Appendix.
7. Consider addressing the following minor issues regarding the paper writing.
   * According to the [resource about how to format the paper](https://github.com/JmlrOrg/tmlr-style-file/archive/refs/heads/main.zip) in the [Author guidelines](https://jmlr.org/tmlr/author-guide.html), we distinguish in-text citations and in-parenthesis citations. In particular,

    > When the authors or the publication are included in the sentence, the citation should not be in parenthesis,

      You should use `citet{}` in this case, and you should use `citep{}` otherwise. I see too many exceptions to this rule. Please fix all the citations properly.

   * Run a spell checker or proofread to catch minor typos, e.g., “generaator”.
   * In **RDI** description (Page 5), "Similarly to SI" should be changed to "Similarly to SIM".
   * In **Relation to Transferability across Images** (Page 21), the paragraph's first sentence is redundant.

**Strengths And Weaknesses:**

# Strengths
1. To the best of my knowledge, this is the first survey on adversarial example transferability.
2. The authors introduce many research works about enhancing adversarial transferability, which will benefit other researchers interested in the transferability of adversarial examples.

# Weaknesses
1. The authors mainly focus on introducing transferability-enhancing techniques. While I’m unaware of all the literature on adversarial transferability, it would be better to include more discussions on theoretical findings or explanations about what causes adversarial transferability.
2. The authors should spend more time proofreading the writing and correcting mistakes in citation, spelling, typos, etc. In particular, typos in mathematical notations must be corrected; a survey paper should convey information as accurately as possible. Please take a look at** Requested Changes** for more detailed comments.

---

> ### Author Response · Authors · 2024-03-20
> **Reponse to review comment**
>
> We sincerely appreciate the positive feedback regarding our survey being the first of its kind and recognizing the significant value it brings to the community. Our response is as follows:
>
> **W1**: It would be better to include more discussions on theoretical findings or explanations about what causes adversarial transferability.
>
> **R1**: We agree with the reviewer that the theoretical analysis and the explanations of adversarial transferability are indeed important. Following the reviewer’s suggestion, we summarize related work from this perspective and add a deep discussion in section 6.1. The takeaway message is that there is still a lack of theoretically well-grounded analysis, while some research attempts to understand the root of adversarial transferability.
>
> **W2**: The authors should spend more time proofreading the writing and correcting mistakes in citation, spelling, typos, etc. In particular, typos in mathematical notations must be corrected;
>
> **R2**:  Thank the reviewer for pointing this out. Following the suggestion, we did proofreading for multiple rounds with the help of native speakers and fixed some typos and grammatical errors. Especially, we double-check the equations by comparing them with the original paper and explain each notation therein.
>
> **Q1**: Consider renaming the two categories of transferability-enhancing methods. ... In Section 3. I’d name this category optimization-based methods.
>
> **A1**: We appreciate the suggested categorization name, which indeed makes the categorization more clear. Specifically, we changed surrogate model-based to optimization-based and generative model-based to generation-based. The sub-categorization of optimization-based are data augmentation, optimization technique, loss objective, and model component. A summary table is presented in table 1. Thanks again for the great suggestion for improving our paper, which benefits a lot to our community.
>
> **Q2**: I can see a few typos in mathematical notations/formulae. I’ll list some of those mistakes, but I cannot guarantee that I covered all the formulae. Please review all the formulae and ensure there are no typos.
>
> **A2**: Thank the reviewer very much for spending so much time and effort to help us with the details. We fixed those errors and typos. Besides, we also review all our equations and annotations to make sure they are correct and improve their readability. Thanks!
>
> **Q3**: Sometimes, the authors use terms or notations without explaining what they mean. Please describe the missing parts.
>
> **A3**: Thanks for the suggestions. We update our annotation table and some equations to make sure they are applied consistently across the paper. Besides, we also add some explanations of the complicated equations and notations therein to improve their readability.
>
> **Q4**: In Table 1, the untargeted attack success is determined by the misclassification of a model. However, Equation 1 uses "change of prediction" as an attack success. Either change the description of the untargeted attack in Table 1 or change Equation 1
>
> **A4**: Thanks for pointing the inconsistency out. We fixed it by changing Equation 1, which is now marked in blue in our paper.
>
> **Q5**: The first paragraph in Section 3.2 contains an interesting perspective. Can we interpret other methods (out of Section 3.2) from this viewpoint, i.e., data augmentation as another approach to enhance the generalization?
>
> **A5**: Thanks for pointing out the connections between different subsections of section 3. In our current categorization, section 3 presents the optimization-based transferability-enhancing method, and each subsection corresponds to the different perspectives of the optimization, such as input data, optimization loss, optimization model, and optimization algorithm itself. We polished the origin arguments in section 3.2 to follow this new categorization.
>
> **Q6**: In Section 3.2, separate the iterative variants of FGSM from the methods for adversarial transferability. You can add a \subsubsection summarizing the variants or move them to other sections, e.g., Appendix.
>
> **A6**: Thanks for the great suggestion, which significantly increases the readability of this section. Following the reviewer’s suggestion, we summarize all the variants of I-FGSM, present one classical variant in the main text, and put more variants with details in the appendix.
>
> **Q7**: Consider addressing the following minor issues regarding the paper writing and citation format.
>
> **A7**: Thank the reviewer for pointing out the correct format for citations. Following the guidance provided by the reviewer, we fixed all the citation format issues. Thanks.
>
> **Q8**: minor issue: Run a spell checker or proofread to catch minor typos, e.g., “generaator”.
> In RDI description (Page 5), "Similarly to SI" should be changed to "Similarly to SIM".
>
> **A8**: Thanks for helping us with this. We fixed them. Besides, we also did grammar checks and proofreading for multiple rounds.

---

### Review · Reviewer_r2cH · 2024-03-11

**Summary Of Contributions:**

This submission is a survey paper. It provides a comprehensive review of research progress on achieving and improving the transferability of adversarial examples. The paper mainly focuses on transferability-based attacks in the image classification task, covers a broad range of these attacks, and classifies these attacks into two categories: via surrogate model, and via generative model. Then the paper further discusses the transferability-based attacks in other domains. The paper is concluded with a discussion of future works.

**Audience:**

Yes

**Broader Impact Concerns:**

The work may need to add a Broader Impact Statement to summarize the potential malicious use of adversarial transferability-based attacks and potential mitigations.

**Claims And Evidence:**

Yes

**Requested Changes:**

See "weaknesses" above for requested changes.

Minor:
- Table 2: define "AE" - I guess it is "adversarial example"?
- Page 3 last line: "pertrubations" typo
- Page 4 above Eqn. (7) "in Equation 6 is often solved" -> "is often approximately solved"
- Page 6: "Lin et al highlights": "Lin et al" is in plural form, so it should be "Lin et al highlight" Same to several other occurrences.
- Page 10: " It is not enough for targeted adversarial examples to be close to the target class but not far from the true class." seems not very clear.
- Eqn. (41): abs R_i seems to be a typo
- Below Eqn. (45), $S_i$ denotes the set of activations. What is "the set of activations"?
- Last line in Page 21, $L - p$ -> $L_p$

**Strengths And Weaknesses:**

Strengths:
- A comprehensive survey covering most impactful and recent works in developing transferabiltiy-based attacks.
- A systematic and clear taxonomy. For example, for attacks via surrogate models, the survey classifies them into three classes: data augmentation, loss, and optimization.

Weaknesses:
- The submission may be improved in terms of summarization. Several parts of the submission read more like pure enumeration (xxx et al present xxx; xxx et al present xxx; ...) instead of categorizing and summarizing the literature. It would benefit the readers more if the core methodology is revealed and used to narrate the literature top-down. Furthermore, some taxonomy figures and comparison tables would improve the accessibility.
- The granularity assignment may be improved. For attacks vis surrogate models, many works are introduced in detail, with core equations listed but without a complete explanation. For example, $m$ and $v$ are not defined in AI-FGTM. Eqns. (27) and (28) are presented without detailed explanation. On the other hand, in Section 5.3, only the goal of the attacks are revealed (present an attack achieving xxx) rather than core methodology. It can be better if the core methodology of the representative methods is introduced and other details are omitted.
- The discussion can be further enhanced. For example, I would expect the authors to contextualize the research of adversarial transferability in a larger scope - their impact and connections with general adversarial ML, the theoretical point of views of adversarial transferability, their applications in more general trustworthy AI, the transferability study prior to the DL era, the transferability study in LLMs, etc. However, the discussion of the submission seems to mainly focus on how to leverage transferability to derive stronger adversarial attacks itself.

---

> ### Author Response · Authors · 2024-03-20
> **Reponse to review comment**
>
> We thank the reviewer for the positive feedback on the literature comprehensiveness of our survey and a systematic and clear taxonomy on the topic. Our response to review comment is as follows:
>
> **Q1**: The submission may be improved in terms of summarization. Several parts of the submission read more like pure enumeration (xxx et al present xxx; xxx et al present xxx; ...) instead of categorizing and summarizing the literature. It would benefit the readers more if the core methodology is revealed and used to narrate the literature top-down. Furthermore, some taxonomy figures and comparison tables would improve the accessibility.
>
> **A1**: Thanks for the great suggestions. Following the suggestions, we first add a table (table 1) to show our taxonomy of adversarial transferability. The table shows our categorization in different levels and citations of related work. Moreover, for each subcategory, we provide a unified mathematical formulation and show how the related work can be expressed as variants of the formulation. Besides, we also add a summary of related work under each subcategory before delving into the details, which is marked in blue across the paper.
>
> **Q2**: The granularity assignment may be improved. For attacks vis surrogate models, many works are introduced in detail, with core equations listed but without a complete explanation. For example, m and v are not defined in AI-FGTM. Eqns. (27) and (28) are presented without detailed explanation. On the other hand, in Section 5.3, only the goal of the attacks are revealed (present an attack achieving xxx) rather than core methodology. It can be better if the core methodology of the representative methods is introduced and other details are omitted.
>
> **A2**: The reviewer for pointing out the uneven granularity assignment. The granularity of our current categorization is assigned as follows: 1). For the transferability in the image classification task (sections 3 and 4), we provide the details of the methods, i.e., their mathematical formulations. The granularity is consistently followed for both optimization-based (surrogate) and generation-based approaches in classification tasks. The core technology of each subcategory is expressed as a unified formulation at the beginning of each subcategory. 2). For the tasks beyond image classification (section 5), we only provide a summary of adversarial transferability in each task. Since there are only a few (or a single) transferability-enhancing methods, detailed mathematical formulation and task introduction would be too much content for this survey. Thus, instead, we provide a brief summary with a new granularity for the ones beyond image classification.
>
> **Q3**: The discussion can be further enhanced. For example, I would expect the authors to contextualize the research of adversarial transferability in a larger scope - their impact and connections with general adversarial ML, the theoretical point of views of adversarial transferability, their applications in more general trustworthy AI, the transferability study prior to the DL era, the transferability study in LLMs, etc. However, the discussion of the submission seems to mainly focus on how to leverage transferability to derive stronger adversarial attacks itself.
>
> **A3**: We appreciate a lot for the suggestion to enrich our discussion. As the reviewer realized, our original submission aims to focus on transferability-enhancing methods. But we totally agree with the reviewer that adding the connections to broader topics will benefit more to the community. Following the reviewer’s suggestions, we add section 6.2 with a focus on connections of adversarial transferability to adversarial ML, trustworthy AI as well as adversarial transferability prior to the deep learning era. Besides, we also add a summary of adversarial transferability across LLMs in section 5.2 (transfer attack in NLP attacks). A discussion about adversarial transferability across large multimodal models is also added in section 6.1. Thank the reviewer for suggesting these innovative points.
>
> **Q4**: Minor issue on typos
>
> **A4**: Thank the reviewer for pointing this out. We proofread our paper for multiple rounds with the help of native speakers and fixed some typos and grammatical errors (the fixed content is marked in blue).
>
> **Q5**: The work may need to add a Broader Impact Statement to summarize the potential malicious use of adversarial transferability-based attacks and potential mitigations.
>
> **A5**: We add a Broader Impact Statement at the end of the paper to discuss the potential impact of adversarial transferability on the community.

---

> > ### Comment · Reviewer_r2cH · 2024-04-16
> >
> > Thanks for the revision. The quality is greatly improved and I don't have other follow-up questions.

---

### Author Response · Authors · 2024-03-20
**General Response to AE and all Reviewers**

We first thank reviewers for their positive feedback on the following points: **regarding our survey being the first** (Reviewer 7rTq), **a systematic and clear taxonomy on the topic** (Reviewer r2cH), **the literature comprehensiveness of our survey** (Reviewer ifNX and 7rTq), **the well-organized paper structure** (Reviewer ifNX), and **the significant value it brings to the community** (Reviewer r2cH and 7rTq).

We also appreciate the reviewers’ suggestions. As a response, we made the main changes as follows:
1. We added a table (table 1) to show our taxonomy of adversarial transferability. The table shows our categorization in different levels and citations of related work.

2. We extended section 6 into multiple subsections. In section 6.1, we discuss more challenges we summarized from the current literature and potential future directions as opportunities. In section 6.2, we add connections of adversarial transferability to adversarial ML, trustworthy AI as well as adversarial transferability prior to the deep learning era.

3. We summarized all the variants of I-FGSM in section 3.2, present one classical variant in the main text, and put more variants with details in the appendix.

4. We summarized related work from this perspective of theoretical analysis and explanations and added a deep discussion in section 6.1.

5. We added a Broader Impact Statement at the end of the paper to discuss the potential impact of adversarial transferability on the community.

6. We proofread our survey paper for multiple rounds and fixed some typos, grammatical errors, citation format as well as other minor issues.

We sincerely thank all of you for your valuable time and insightful suggestions for improving our paper.

---

> ### Author Response · Authors · 2024-03-23
> **A friendly reminder that our discussion period ends soon**
>
> Thank all reviewers again for their valuable time spent on our work and insightful suggestions.
>
> This is a friendly reminder. Our discussion period ends soon in two days. Please let us know if you have more suggestions.
>
> We are happy to further improve our paper if reviewers have more suggestions for our updated version.
>
> Best Regards!
> Authors

---

### Public Comment · ~Lai_Cheng_Lin1 · 2024-03-21
**Section 3.1 SIM method is not resize**

I think in section 3.1, the SIM method is not resize operation. It's the scaling of pixel value not resizing.

Refer to "Wang, X., He, X., Wang, J., & He, K. (2021). Admix: Enhancing the transferability of adversarial attacks. In Proceedings of the IEEE/CVF International Conference on Computer Vision (pp. 16158-16167)." The author said that admix is a generalization of SIM, while admix is doing interpolation between pixel, so
 the scaling is multiplying pixel value by a constant instead of resize operation.

---

> ### Author Response · Authors · 2024-03-21
> **Thank you for your interest and suggestion.**
>
> Dear Lai,
>
> Thank you for your interest in reading your paper.
>
> Scale-Invariance Method aims to build different scale copies of given input images. The operation presented in the original SIM paper is indeed the pixel scaling operation, as you pointed out. Other operations are inappropriate to be stated as a representative although they are possible. Thus, we modify it to pixel scaling operation.
>
> Thank you for pointing out the difference.
>
> Best Regard! Authors

---

### Decision · Action_Editor_1QyD · 2024-04-22

**Recommendation:** Accept as is

**Comment:**

While the paper does not offer any new or novel approaches to adversarial transferability, the consensus is that the paper provides an extensive literature review and will benefit future research. Concerns were raised regarding the comprehensiveness, depth of analysis and comparison, and presentation quality. These concerns were addressed by the authors during the discussion phase, significantly enhancing the paper's quality.

**Audience:**

I believe that, aligned with the reviewers' recommendations, the paper will be helpful for readers who want to dive into research on adversarial transferability.

**Claims And Evidence:**

A comprehensive review and categorization of the literature on adversarial transferability is shown in the paper.